

**Photoacoustic measurement may significantly overestimate NH₃ emissions from cattle**
**houses due to VOC interferences**
Dezhao Liu[1,2*], Li Rong[2], Jesper Kamp[2], Xianwang Kong[1], Anders Peter Adamsen[3], Albarune
Chowdhury[2], Anders Feilberg[2*]
1-  Zhejiang University, College of Biosystems Engineering and Food Science, Yuhangtang

9        Road 866, 310058 Hangzhou, China

2-  Aarhus University, Department of Engineering, Finlandsgade 22, 8200 Aarhus N, Denmark
3-  APSA, c/o Agro Business Park, Niels Pedersens Allé 2, DK-8830 Tjele, Denmark
**Corresponding author: Dezhao Liu: dezhaoliu@zju.edu.cn;**
**Anders Feilberg: af@eng.au.dk**



**Abstract**: Infrared photoacoustic spectroscopy (PAS) is a widely used method for measurement
of $NH_3$ and greenhouse gas emissions especially in agriculture, but non-targeted gases such as
volatile organic compounds (VOCs) from cattle barns may interfere with target gases causing
inaccurate results. This study made an estimation of $NH_3$ interference in PAS caused by selected
non-targeted VOCs which were simultaneously measured by a PAS and a PTR-MS (proton
transfer reaction mass spectrometry). Laboratory calibration were performed for $NH_3$
measurement and VOCs were selected based on a headspace test of the feeding material maize
silage. Various levels of interference of tested VOCs were observed on $NH_3$ and greenhouse
emissions measured by the PAS. Particularly, ethanol, methanol, 1-butanol, 1-propanol and
acetic acid were found to have highest interference on $NH_3$. A linear response was typically
obtained, with non-linear relation was however observed for VOCs on $N_2O$ emissions. The
corrected online $NH_3$ concentrations measured by the PAS from a field study were confirmed
to be reasonably correlated to the $NH_3$ concentration measured simultaneously by the PTR-MS.
It was concluded that the correction factors could be used for possible data corrections when
the concentrations of VOCs could be obtained by using e.g. PTR-MS.

## 1  Introduction

Measurements of ammonia and greenhouse emissions are gaining increased research attention
in recent years due to stronger interests on global change and air pollution. Especially, ammonia
not only causes serious environmental problems such as soil acidification as well as pollution
of underground water and surface water with nitrogen eutrophication (van Breemen et al., 1983;
Pearson and Stewart, 1993; Erisman et al., 2007), but is also important for fine particle



formation (Bouwman et al., 1997; Seinfeld and Pandis, 1997; Pinder et al., 2007). The
greenhouse gas emissions, on the other hand, are causing climate change (Thomas et al., 2004;
Chadwick et al., 2011). Livestock husbandry was estimated to be responsible for more than 80 %
of the ammonia emission in Western Europe (Hutchings et al., 2001; EMEP, 2013) and more
than 60% in China (Paulot et al., 2014). In the U.S., agriculture accounts for ~90 % of the total
ammonia emissions (Aneja et al., 2009). Meanwhile, agriculture accounts for 52 and 84 % of
global anthropogenic methane and nitrous oxide emissions (Smith et al., 2008). Accurate
measurements of ammonia and greenhouse emissions are therefore vital for reliable emission
estimation and thereby the possible reduction of these emissions through various efforts, such
as air cleaning with biotrickling filters and air scrubbers (Melse and Van der werf, 2005; De
Vries and Melse, 2017). For ammonia measurements, more than 30 % difference was observed
when various methods were compared (Scholtens et al., 2004).
Infrared photoacoustic spectroscopy (PAS) is a widely-used technique for studies of air
emissions especially within agriculture (Osada et al., 1998; Osada and Fukumoto, 2001;
Emmenegger et al., 2004; Schilt et al., 2004; Heber et al., 2006; Elia et al., 2006; Blanes-Vidal
et al., 2007; Hassouna et al., 2008; Rong et al., 2009; Ngwabie et al., 2011; Cortus et al., 2012;
Joo et al., 2013; Wang-Li et al., 2013; Iqbal et al., 2013; Zhao et al., 2016; Ni et al., 2017; Lin
et al., 2017). The PAS technique determines the gas concentrations through measuring acoustic
signals caused by cell pressure change when gas inside absorbs energy from infrared light at a
specific wavelength using the optical filter and expands (Iqbal et al., 2013). For example, the
Innova 1312 (AirTech Instruments, Ballerup, Denmark) uses the PAS method and was
previously verified by the US EPA and recommended by the Air Resources Board in California



(CARB, 2000). Besides, PAS has the advantages of performing continuous measurement with
low maintenance and good selectivity and can simultaneously measure five compounds,
typically including $NH_3$, $CH_4$, $CO_2$ and $N_2O$ for agricultural applications. The water vapor was
usually also included in order to make proper concentration corrections when necessary.
Nevertheless, since the infrared spectroscopic method is applied for measuring gas
concentrations in PAS, the overlapping of IR spectra with non-targeted gases can introduce
significant interferences due to the adsorption of infrared light at similar wavelengths, even
though the infrared bands selected by optical filters are relatively narrow. The interferences can
be corrected through cross-compensation for all target gases when the instrument is calibrated
(Lumasense, 2012), but understanding and estimation of interferences from possible non-
targeted gases is very important. This is especially important for field applications where the
manure or the animal feed may emit various types of gases depending on the management and
operations in the animal houses (Hafner et al., 2010; Moset et al., 2012). Until now, the PAS
interference of has not been well estimated and corrected for, although interferences were
previously suspected in livestock facilities (Phillips et al., 2001; Mathot et al., 2007; Ni & Heber,
2008). Flechard et al. (2005) suspected that the $N_2O$ concentration from soil measured by PAS
(Innova 1312) was heavily influenced by $CO_2$ and temperature even when cross-interference
compensation was applied; they developed an alternative correction algorithm based on
controlled $N_2O/CO_2/H_2O$ ratios under selected temperature. Zhao et al (2012) claimed that the
internal cross compensation could eliminate the interferences between target gases, and
quantified interferences of non-targeted gas of $NH_3$ on targeted gases of ethanol, methanol, $N_2O$,
$CO_2$, and $CH_4$, however, without giving specific correction factors. Iqbal et al. (2013) also





demonstrated that a careful calibration could eliminate the internal cross interferences of high
water vapor and $CO_2$ concentrations on low concentrations of $N_2O$ at the soil surface by
comparison to GC measurements. Nevertheless, tests on interferences by non-targeted VOCs
were not included in their study, likely due to the typical low concentrations of VOC in soil
(Insam and Seewald, 2010). Hassouna et al. (2013) presented a field study on dairy cow
buildings, where interferences on $NH_3$, $CH_4$ and $N_2O$ were observed. The interferences were
suspected to be caused by volatile organic compounds (VOCs; acetic acid, ethanol and 1-
propanol) that they measured simultaneously; two PAS instruments were applied with one of
them allocated with optical filters of these VOCs ($NH_3$ optical filter was included for both PAS).
Still, no correction factors were given in terms of tested volatile organic compounds, which
were typically emitted from feeding materials such as maize silage (Howard et al., 2010;
Malkina et al., 2011). Opposite to what was claimed by some previous studies (e.g., Heyden et
al., 2016), the correction of interferences of non-targeted VOCs on $NH_3$ emission is also
essential for the evaluation of emission abatement technologies such as air scrubbers, especially
when the inlet VOC concentrations are relatively high. An overestimation of ammonia removal
efficiency could easily be obtained since less interference would be expected for the outlet
VOCs especially for water-soluble compounds such as the VOCs investigated in this study.
This study, therefore, performed an evaluation on ammonia measurements and interferences by
non-targeted gases of volatile organic compound on targeted $NH_3$ and greenhouse gases
measurement by PAS, with the interference on $NH_3$ simultaneously demonstrated by Proton-
transfer-reaction mass spectrometry (PTR-MS), Cavity Ring-Down Spectroscopy (CRDS) and
PAS. The experiments were as follows: (1) ammonia laboratory calibration by the three



instruments of PAS, PTR-MS and CRDS; (2) VOC selection test for non-targeted interference
to ammonia by the PAS; (3) Effect of non-targeted VOCs on ammonia and greenhouse
emissions measured by the PAS; (4) Field confirmation of interferences of non-targeted VOCs
on ammonia measurement and data correction.

**2    Materials and methods**
**2.1    Instrumentation for gas concentrations measurement**
In this study, a PTR-MS, a CRDS $NH_3$ analyzer and a PAS gas analyzer were used to measure
gas concentrations. PTR-MS is a state-of-the-art and widely used CIMS (short for chemical-
ionization mass spectrometry) technique for highly sensitive online measurements of VOCs
(De Gouw and Warneke, 2007; Blake et al., 2009; Yuan et al., 2017). PTR-MS can also measure
a few inorganic compounds such as ammonia (at m/z 18) since the proton affinity (204.0
kcal/mol) of ammonia is higher than that of water (165.0 kcal/mol). Due to the fact that intrinsic
ion at m/z 18 could be formed in the plasma ion source (Norman et al., 2007), ammonia
measurement by PTR-MS need to be evaluated carefully. For agricultural applications with
relative high ammonia concentrations (e.g., Rong et al., 2009), this high background is usually
not a big problem, since the typical background concentration is only a few hundred ppbv.
When total gas concentration measured by PTR-MS is higher than approximately 10 ppmv,
dilution is needed to keep the stable level of primary ion signals. A high-sensitivity PTR-MS
(Ionicon Analytik GmbH, Innsbruck, Austria) was applied for the test of ammonia calibration
in the laboratory, effects of non-targeted VOCs on ammonia measurement and field
confirmation of interferences of non-targeted VOCs on ammonia measurement. Standard





133 conditions with a total voltage of 600 V in the drift tube were utilized for the PTR-MS. Pressure

134 and temperature in the drift tube were maintained in the range of 2.1-2.2 mbar and at 60 °C,

135 respectively, which gives an E/N ratio of ca. 135 Townsend. The inlet of the PTR-MS is PEEK

136 tubing of 1.2 m length with 0.64 mm inner diameter (ID) and 1.6 mm outer diameter (OD). The

137 inlet flow to the PTR-MS during calibration test and measurements was kept ~150 mL/min.

138 The inlet temperature was maintained at 60 °C. Mass calibration was performed before each

139 test, while transmission calibration was performed for every two weeks as suggested by the

140 manufacturer.

141 CRDS determines the gas concentration (e.g., $NH_3$) by measuring the acceleration of ring down

142 time of light in the cavity due to absorption by a targeted gas species, this is compared to the

143 'normal' ring down time of the light introduced by a laser with tunable wavelength (von

144 Bobrutzki et al., 2010; Picarro, 2017). The very long effective path length of the light in the

145 cavity (e.g., over 20 km for 25 cm cavity) (Picarro, 2017), enables a significantly higher

146 sensitivity compared to conventional absorption spectroscopy (Berden et al., 2000; von

147 Bobrutzki et al., 2010). A G2103 Analyzer (Picarro Inc., Sunnyvale, CA, USA) using CRDS

148 technique was applied in this study for the test of ammonia laboratory calibration and for the

149 effect of non-targeted VOCs on ammonia measurement. The manufacturer calibrated the CRDS

150 analyzer approximately 3 months before calibration tests and interference measurements. The

151 CRDS analyzer was equipped with two in-line, sub-micron polytetrafluoroethylene (PTFE)

152 particulate matter filters; one at the gas inlet at the back of the analyzer and one at the inlet of

153 the cavity to protect the highly reflective mirrors. The inlet of the CRDS is a Teflon (PTFE)

154 tubing of 1.5 m length with 6.4 mm outer diameter. Since molecular spectroscopy is



fundamentally affected by temperature and pressure, the CRDS's optical cavities incorporate
precise temperature and pressure control systems, with the measurement cell temperature
controlled under precision of $\pm 0.005$ °C, while the measurement cell pressure controlled
under precision of $\pm 0.0002$ atm. In this study, both the temperature and pressure of the air
sample continuously flowing through the optical cavity are tightly controlled at all times to
constant values of 45 °C and 140 Torr, respectively. The measurement interval is around 3
seconds. The CRDS analyzer measured the water vapor simultaneously.
A photoacoustic multi-gas monitor 1312 (Innova, Lumasense Technology A/S, Denmark) using
PAS technique was compared with the PTR-MS and the CRDS for ammonia calibration and
non-targeted VOCs on ammonia measurement. The sample integration time to measure
ammonia by the PAS was 20 s. The PAS used 6 optical filters including $NH_3$, $CH_4$, $CO_2$, $H_2O$,
$N_2O$ and $SF_6$. The specifications of the optical filters are shown in Table S1. Water vapor must
be included for PAS measurement, since the absorbance spectrum of water overlap with other
gases such as $N_2O$ and $CO_2$ thus causing interferences. The supplier calibrated the PAS before
the conduction of the measurements for comparison in this study. The interferences between
the target gases were therefore supposed to be eliminated through internal cross compensation
(Lumasense, 2012; Zhao et al., 2012).
**2.2 Experiment 1:laboratory test on ammonia calibration**
The background measurement, calibration on selected ammonia concentrations, and reaction
time and decay time measurement were performed for ammonia measurement by PAS, PTR-
MS and CRDS. For the background measurement, zero air controlled by a mass flow controller
(Bronkhorst, Ruurlo, The Netherlands) was supplied, and measurement was performed



individually for each instrument. The selected ions measurement mode was used for the PTR-
MS with m/z 18 being used for ammonia measurement. For the calibration test, a factory-
calibrated gas cylinder (AGA A/S, Copenhagen, Denmark) containing 99.7 (± 10 %) ppmv
ammonia was used for the calibration test. Mass flow controllers (Bronkhorst, Ruurlo, The
Netherlands) were used to dilute the cylinder gas with zero air to achieve the desired $NH_3$
concentration levels. For the decay time test, zero air flow was supplied to the instruments at
first, then switched to a diluted flow (via 2-levels of mass flow controllers) with ammonia
concentration around 5.2 ppmv supplying to all three instruments simultaneously, afterwards
the ammonia supply flow was then set to zero to test the decay time. Four individual decay time
tests were performed for the PAS, in order to confirm the long decay time of the instrument
with low ammonia concentrations (5.2-8.8 ppmv) or high ammonia concentration (99.7 ppmv).
For the reaction time test for the PAS, two different levels of ammonia concentration were
introduced individually to the instrument, in order to test the dependence of the reaction time
on ammonia concentration.
**2.3   Experiment 2: VOCs selection test**
In order to prepare the interferences test of non-targeted VOCs on ammonia measured by the
PAS, a headspace test was performed and VOCs were selected through a PTR-MS measurement.
Maize silage is a typical feeding material to the cows. A sample of maize silage was collected
from the farm where the field confirmation experiment was performed (Skjern, Jutland,
Denmark, altitude: 55°59′36.6″, longitude: 8°29′53.52″). The silage was then transferred to the
laboratory immediately for the headspace test. A clean plastic container (58×38×43 cm) with
two oval holding holes on sides was used for the headspace test for VOCs selection. The



container was half opened and the silage filled half of the container. A 1-meter 1/4-inch ID
PTFE tube was used for the test, with one end placed around 5 cm above the silage, and the
other side connected to a T-piece. One side of the T-piece was connected to a 1/8-inch ID PTFE
tube (around a half meter) which is connecting to the inlet of the PTR-MS. The flow rate of the
PTR-MS was kept at 150 mL/min. A zero-air dilution flow (75 mL/min) was supplied to the T-
piece in order to make 1:1 dilution to keep the total concentration below 10 ppmv. The
headspace measurement was performed by the PTR-MS on scan mode, and masses were
measured from 21 to 250 with 200 ms for each mass. The selection of VOCs was based on the
scan results and relevant literature for silage (Howard et al., 2010; Malkina et al., 2011).

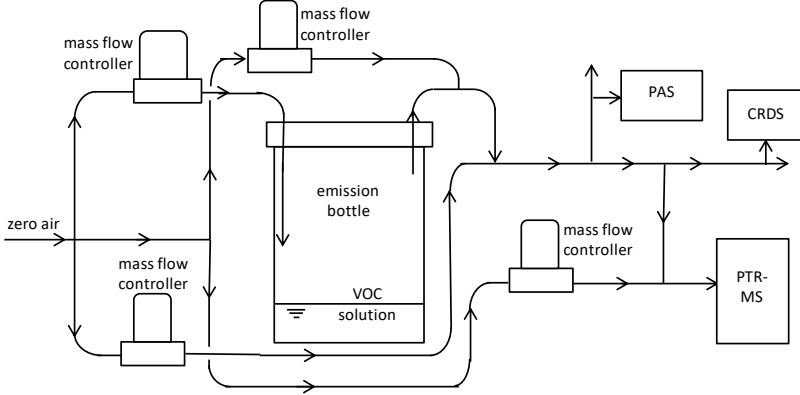


**Figure 1.** The diagram of experimental set-up for ammonia interference calibration from VOCs.
**2.4   Experiment 3: Laboratory test for correction factors**
The diagram of the setup for the laboratory calibration test is shown in Figure 1. In the setup, a
pre-tested water solution containing the single VOC was purged from the headspace by zero air
(or nitrogen for one test on methanol), with flow controlled by a mass flow controller. The flow
was set with care, due to the relatively high sensitivity of VOC concentration on the purged gas



flow rate. One-liter airtight glass bottles were used for holding the water solution containing
the VOC, and 1/4-inch ID PTFE tube was used for the pipelines in the setup. The purged air
flow in the PTFE tube containing a single VOC was diluted with air through a two-step dilution.
The flows were adjusted according to the purged VOC concentration and the desired final VOC
concentration. The pre-test for water solution preparation used a ratio of VOC:Water as 1:5,
and the ratio between VOC and water was adjusted if the purged concentration after dilution
(by zero air controlled by 2 mass flow controllers) measured by the PTR-MS was not within
the desired range (too low or too high). For the laboratory calibration test, the diluted VOC was
connected to the PAS, the CRDS and the PTR-MS for simultaneous measurements. In order to
maintain stable pressures in the PAS and the CRDS, specific ranges of excess flow rates were
required for these two instruments. Specific, the excess flow for the PAS was kept around 4 L
min$^{-1}$, while the excess flow for the CRDS was kept around 2 L min$^{-1}$. For the PTR-MS
measurement, a further dilution by zero air was typically used to keep the total concentrations
below 10 ppmv in order to avoid depletion of the primary ion, $H_3O^+$. Selected ion measurement
mode was applied for the PTR-MS, with an integration time of 2 seconds for the tested VOC
mass. During the experiments, the humidity was kept relatively low and stable, with dry zero
air used for dilution for all cases, except for one test on methanol, which was also tested under
nitrogen condition.
**2.5   Experiment 4: Field test for validation of correction factors**
The field demonstration test for non-targeted VOCs on ammonia measurement by the PAS was
performed in the dairy farm mentioned above (Skjern, Jutland, Denmark), where both the PTR-
MS and the PAS measured continuously. The dairy farm housed 360 cows with an average
weight of 650 kg. The ventilation system consisted of natural and mechanical partial pit
ventilation system.
For the field test, the PAS was combined with a Multiplexer 1309 to measure from several
sampling points. The PAS and the PTR-MS were placed in a movable trailer next to the dairy
farm. The manufacturer calibrated the PAS instrument before the field test. The sample
integration time was 5 s and the flushing time was 20 s. The air concentrations were measured
by the PAS sequentially among two selected locations inside the farm, one location in the pit
ventilation, one location outside the farm. Teflon tubes of 20 meters long and 8 mm OD were
used for the sampling of air. The sampling lines were connected with the channels of the PAS
multi-point sampler via continuously running Teflon membrane pumps to ensure constant
flushing. Selected VOCs, odorants and $NH_3$ were measured simultaneously by the high
sensitivity PTR-MS. Measurements were switched among the four measurement sampling lines
and the background at ca. 10 min intervals via a custom-built switching box. PTFE tubes were
used for the PTR-MS sampling lines, which were connected to Teflon sampling lines before the
Teflon membranes pumps. The switching box was equipped with a five-port channel selector
(Bio-Chem Valve Inc, USA) controlled automatically by 24V outputs from the PTR-MS. A
PTFE tube (ID 1 mm) was used to connect the switching box to the inlet sampling line (1-meter
PEEK tube with ID 0.64 mm) of the PTR-MS. For selected compounds, calibration was
performed for the PTR-MS before the field measurements using permeation tubes and reference
gas mixtures. Details regarding the calibration procedures could be found in our previous study
(Feilberg et al., 2010). Standard conditions as described previously was applied and maintained
for the PTR-MS. The mass discrimination was calibrated and adjusted weekly by using a





mixture of 14 aromatic compounds between m/z (mass to charge ratio) 79 and 181 (P/N 34423-
PI, Restek, Bellefonte, PA). Selected ions were monitored with dwell time between 200 and
2000 ms during each measurement cycle. Masses and dwell time selection was based on ion
abundance in full scan mode, relevant literature and experience regarding odorant compounds
from dairy buildings as well as from pig houses and pig slurry applications (Shaw et al., 2007;
Chung et al., 2009; Liu et al., 2014; Liu et al., 2018).

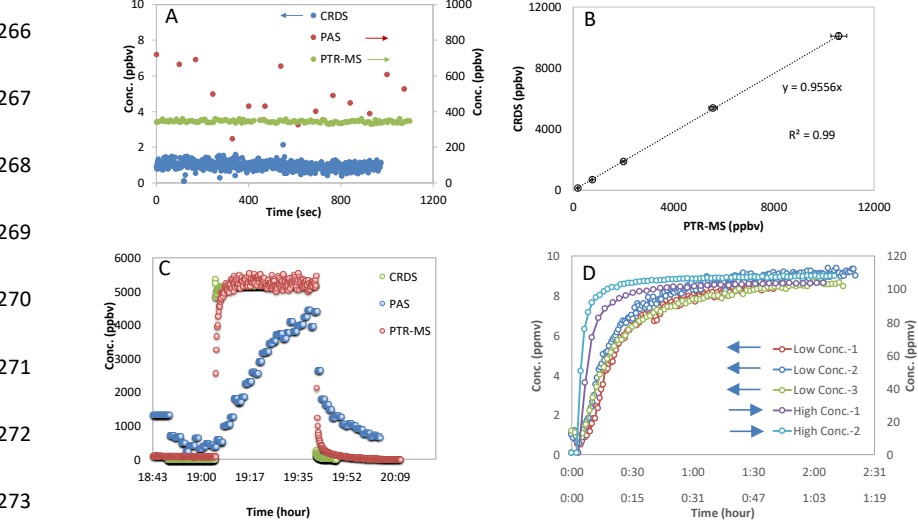

**Figure 2.** The calibration test of ammonia by the PAS, the PTR-MS and the CRDS. A: Background
comparison for the CRDS, the PAS and the PTR-MS for ammonia measurement; B: The calibration
of ammonia measured by the PTR-MS and by the CRDS; C: The instrument decay time of measured
ammonia concentration by the PTR-MS, the PAS and the CRDS; D: The reaction time for ammonia
for the PAS under low concentration (3 tests; ~8.9 ppmv) and high concentration (2 tests; 99.7 ppmv)
conditions (Low Conc.-1, Low Conc.-2 and Low Conc.-3 point to the vertical axis on the left, and
to the upper horizontal axis; High Conc.-1 and High Conc.-2 point to the vertical axis on the right,
and to the lower horizontal axis; High Conc.-2 was tested without the multiplexer).

## 3    Results and discussion

### 3.1    Experiment 1:laboratory test on ammonia calibration

The background concentrations of ammonia measured by PAS, CRDS and PTR-MS,





respectively, are shown in Figure 2A, in which very low background concentration was
observed for the CRDS instrument (around 1 ppbv; 5 s) with detection limit around 0.67 ppbv
(3 times the standard deviation of the background). The PTR-MS, on the other hand, gave much
higher background with nearly 400 ppbv observed. The high background for ammonia
measured from the PTR-MS is caused by the intrinsic formation of $NH_4^+$ (m/z 18) in the ion
source (Norman et al., 2007). Nevertheless, the measured background signals for ammonia by
the PTR-MS was very stable and could be subtracted to give a detection limit of 21 ppbv (3
times the standard deviation of the background). Among the three instruments, the PAS gave
the highest background signal for ammonia (corresponding to $502 \pm 140$ ppb), with a detection
limit around 421 ppbv (3 times the standard deviation of the background).
For the calibration test of ammonia, the ammonia concentrations measured by the CRDS and
the PTR-MS is shown in Figure 2B, in which the linearity (k = 0.9556) and high correlation
($R^2$=0.999) are satisfactory for both instruments. The measured ammonia concentrations also
agreed with expected ammonia concentrations from the ammonia gas cylinder diluted in zero
air.
**Table 1.** Instrumental decay time (in second).

| Unit(s) | PTR-MS (5.2 ppm) | Picarro (5.2 ppm) | Innova (5.2-8.8 ppm) | Innova (100 ppm) |
|---|---|---|---|---|
| 90% decay | 70-80 | 4.5-4.7 | 1700-4000 | 450-550 |


For the decay time test, the instrument decay times for ammonia measurements by the PAS, the
CRDS and the PTR-MS were measured simultaneously under a calibrated ammonia
concentration of 5.2 ppmv. As shown in Figure 2C, ammonia measured by the CRDS showed
the shortest decay time while the PAS gave the longest decay time. The estimated decay time



is shown in Table 1, in which the 90% decay time for ammonia measured by the CRDS is
around 4.5 - 4.7 second, with the 90% decay time from the PTR-MS estimated to be 70 to 80
seconds. The decay time for ammonia measured by the PAS showed remarkably longer, with
estimated 90% decay time around 1700 to 4000 seconds (for four individual tests with ammonia
concentration ranged from 5.2 to 8.8 ppmv). When much higher ammonia concentration was
used (99.7 ppmv), the 90% decay times measured by the PAS were apparently shorter (450 to
550 seconds). This result is consistent with the reaction time tests under two levels of input
ammonia concentrations (~ 8.9 ppmv and 99.7 ppmv, respectively), with the reaction time
comparably much shorter when input ammonia concentration is higher, as shown in Figure 2D.
Besides, the multiplexer attached to the PAS seemed to increase the reaction time, as also shown
in Figure 2D. However, a very high concentration of about 100 ppm is not expected to be
commonly seen in agricultural applications.

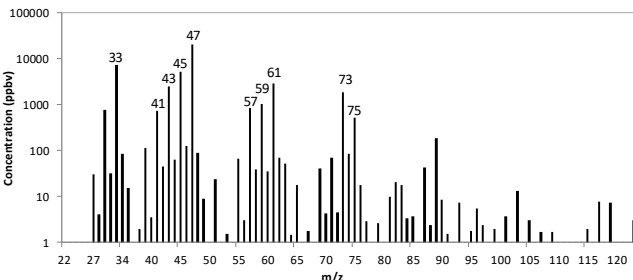


**Figure 3.** A scan example of the feeding material of silage by using headspace technique measured by
the PTR-MS. The m/z 47 is corrected for ethanol fragmentations formed in the PTR-MS through
calibration. Selected VOCs for the test in this study were ethanol, methanol, acetaldehyde, acetic acid,
2-butanone, acetone, propanol and butanol.

**3.2    Experiment 2: VOCs selection test**
The tested VOCs were selected according to a scan test of the headspace from the feeding
material of maize silage performed by the PTR-MS, as shown in Figure 3. The concentrations



shown in the figure were corrected for dilution, while the concentration of mass 47 was
corrected also from the calibration factor by assuming the mass 47 was assigned to ethanol.
Due to the fragmentation of ethanol in the PTR-MS measurement, only a fraction of ethanol
could be detected on mass 47 (Aprea et al., 2007). The highest peaks of the scan were at masses:
47, 33, 45, 61, 43, 73, 59, 75, 57 and 41. From the VOCs typically found in the highest
concentrations in barns and feeding material (Shaw et al., 2007; Chung et al., 2009; Howard
et al., 2010; Malkina et al., 2011; Hafner et al., 2013) and the scan results, a list of VOCs were
selected. The following VOCs were selected for the interferences tests of non-targeted VOC on
ammonia measurement by the PAS: ethanol, methanol, acetaldehyde, acetic acid, 2-butanone,
acetone, 1-propanol and 1-butanol. Compounds such as ethanol, methanol, acetic acid and 1-
propanol were typically found in cattle barns and feeding materials in high concentrations
(Shaw et al., 2007; Ngwabie et al., 2008; Howard et al., 2010; Hafner et al., 2013).



















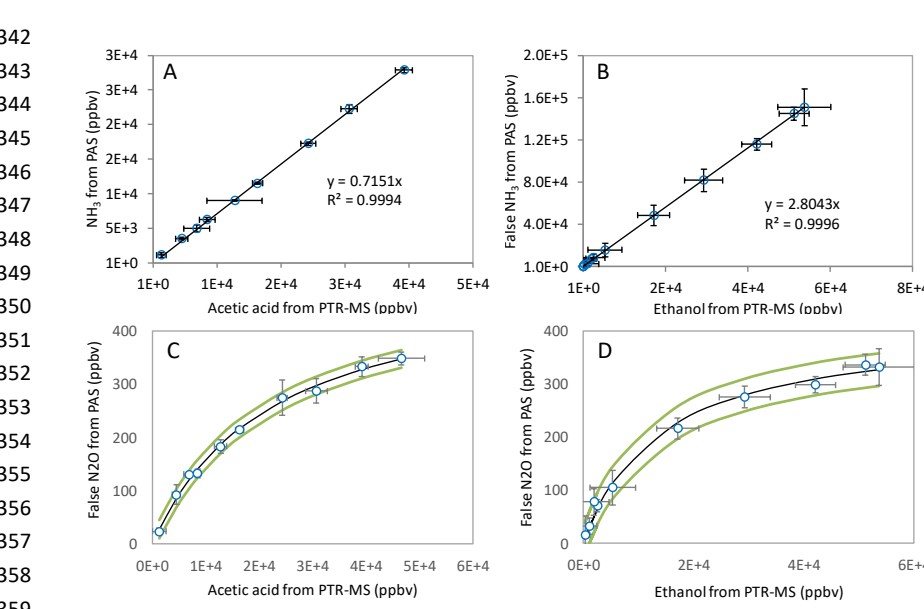



**Figure 4.** Examples for the interference calibration from non-targeted VOC on NH₃ (A & B) and
N₂O (C & D) measured by the PAS. The VOC concentration on horizontal axis was measured by
the PTR-MS, while the NH₃ and N₂O concentrations on vertical axis were from false signals
measured meanwhile by the PAS. A: The interference calibration for acetic acid on NH₃; B: The
interference calibration for ethanol (corrected for fragments through calibration) on NH₃; C: The
interference calibration for ethanol (corrected for fragments through calibration) on N₂O; D: The
interference calibration for acetic acid on N₂O. In C & D, the red line indicated the fit curve by
equation $y=kx/(x+m)$, and the green and purple curves indicated 95% confidence range.
### 3.3 Experiment 3: Laboratory test for correction factors
The interference of non-targeted VOC on ammonia measurement by the PAS was investigated
through selected single VOC as the sole input measured simultaneously by PAS, PTR-MS and
CRDS, as shown in the setup in Figure 1. An example of the interference test can be seen in
Figure S1, where the three instruments measured acetic acid simultaneously under various
concentration levels. Concentration dependent interference was clear for acetic acid on PAS
ammonia measurements.
**Table 2.** Obtained correction factors (in equations) between tested non-targeted VOC and the false
signal measured by PAS. 'y' points to the false concentration measured by PAS, and 'x' points to
the VOC concentration. The value in the brackets indicated the correlation coefficient of the linear
fit. N is the number of VOC concentration levels tested for determination of correction factors.

| Compound | N | NH₃ (y: ppbv; x: ppbv) | CH₄ (y: ppbv; x: ppbv) | N₂O (y: ppbv; x: ppmv) | CO₂ (y: ppbv; x: ppbv) | SF₆ (y: ppbv; x: ppbv) |
|---|---|---|---|---|---|---|
| ethanol | 10 | y=2.81x(1.00) | y=1.88x(1.00) | y=411x/(x+14)(0.93) | y=0.40x(0.99) | y=-0.014x(1.00) |
| methanol | 9 | y=3.29x(0.74) | y=3.81x(0.74) | y=99x/(x+9)(0.78) | y=0.45x(0.47) | y=-0.15x(0.73) |
| acetic acid | 10 | y=0.72x(1.00) | y=-3.14x(1.00) | y=514x/(x+22)(0.95) | y=0.39x(0.99) | y=0.31x(1.00) |
| acetaldehyde | 4 | (-) | y=-0.85x(0.61) | y=317x/(x+31)(0.98) | (-) | y=0.044x(0.58) |
| 2-butanone | 4 | y=-0.13x(1.00) | y=-4.02x(1.00) | y=311x/(x+26)(1.00) | y=-0.61x(0.74) | y=0.23x(1.00) |
| acetone | 6 | y=0.02x(0.99) | y=2.10x(0.99) | y=104x/(x+4)(0.99) | (-) | y=0.015x(0.99) |
| 1-propanol | 5 | y=2.41x(0.87) | y=2.95x(0.87) | y=3569x/(x+602)(1.00) | y=0.25x(0.51) | y=-0.064x(0.84) |
| 1-butanol | 7 | y=2.66x(0.99) | y=3.07x(0.99) | y=807x/(x+73)(0.99) | (-) | y=-0.061x(0.97) |
| methanol(N₂) | 4 | y=1.03x(0.80) | y=1.46x(0.83) | (-) | y=0.35x(0.54) | y=-0.056x(0.86) |



In principle, establishing correction factors for each specific compound could eliminate the
interferences of VOCs on ammonia measurements on a specific instrument with the same filter
specifications. This requires, however, that VOC concentrations be measured simultaneously.



Figure 4A & B show two examples of the calibration lines for acetic acid and ethanol, from
which a correction factor (CF) between the false ammonia concentration and the tested
compound could be obtained (CF=0.72 for acetic acid and CF=2.81 for ethanol). A linear
response of the ammonia interference was observed for all the tested compounds and they had
high correlation coefficients. The correction factors for ammonia interference by other tested
VOCs can be found in Table 2, where ethanol, methanol, 1-propanol and 1-butanol give the
highest false signals on ammonia measured by the PAS, with correction factors of 2.81, 3.29,
2.41 and 2.66, respectively. Due to the fact that these compounds are often found in cattle barn
buildings and feed silage even in the level of ppmv especially for ethanol, methanol and 1-
propanol (Rabaud et al., 2003; Ngwabie et al., 2008; Howard et al., 2010; Hafner et al., 2013),
severe interference on ammonia measured by PAS could therefore exist. While acetic acid gave
significant false signals on ammonia (CF=0.72), acetone only showed little interference on
ammonia (CF=0.02). Meanwhile, negative false signals were observed for ammonia by 2-
butanone (CF=-0.13). Interestingly, the correction factor for false ammonia by methanol in
nitrogen matrix is significantly different from that by methanol presented in air matrix
(CF=1.03 vs 3.29). This observation is possibly related to the relatively rapid vibrational energy
transfer between the VOC and oxygen (Harren et al., 2000). While nitrogen has a vibrational
frequency around 2360 cm$^{-1}$, oxygen has a vibrational frequency of 1554 cm$^{-1}$ with only 170
collisions needed to transfer energy to the vibrational mode of O$_2$ (Lambert, 1977).
Besides the interferences on ammonia by the non-targeted VOCs, other target gases also
showed various levels of interferences, as also indicated by previous studies (e.g., Zhao et al.,
2012; Hassouna et al., 2013). Because target gases may have more overlap for the infrared



spectrum, the primary interference on one target gas caused by the overlap with non-targeted
VOCs could therefore influence and cause secondary interference on other target gases (Zhao
et al., 2012). Still, in theory, correction factors could be obtained for the interfered gases by the
tested VOCs. Specifically, for the interference on methane by non-targeted methanol, 1-butanol,
1-propanol, acetone and ethanol showed positive false signals (CF=3.81, 3.07, 2.95, 2.10, 1.88,
respectively). 2-butanone, acetic acid and acetaldehyde showed negative false signals to
methane, with correction factors equal to -4.02, -3.14 and -0.85, respectively. All interferences
on methane are shown in Table 2. For methanol in nitrogen, the calibration again showed
significant difference compared to air (CF=1.46 vs. 3.81).
Meanwhile, the non-targeted VOC also caused false signals on nitrous oxide signals, with a
much lower level of interference. Further, the calibrations of the nitrous oxide interference by
the non-targeted VOCs seemed not to be following linear relationships. For examples, Figure
4C & D showed the false signals of nitrous oxide caused by ethanol and acetic acid. Clearly, a
non-linear relation exists between the nitrous oxide interference and VOC concentration. The
curves could be well fitted to the non-linear equation of $y=kx/(x+m)$, where k could represent
the maximum interference on nitrous oxide by the single VOC, m could represent the half-
saturation constant indicating the level higher than which of the VOC concentration could cause
half of the maximum interference on nitrous oxide. As shown in Table 2, all tested VOCs
showed positive non-linear interference to the nitrous oxide signals, and 1-butanol showed the
highest maximum interference on nitrous oxide. Interestingly, no interference was observed for
nitrous oxide when methanol was presented in nitrogen matrix, while a relatively lower level
of interference observed on nitrous oxide by methanol when presented in air matrix compared





to other tested VOC.
Furthermore, some of the tested non-targeted VOCs also caused interference on carbon dioxide
measured by the PAS. The background of carbon dioxide was considered as unchanged during
the interference tests. While methanol, ethanol, acetic acid and 1-propanol caused positive false
signals for carbon dioxide measured by the PAS (CF = 0.45, 0.40, 0.39, 0.25, respectively), 2-
butanone caused negative false signals with CF = -0.61 (Table 2). Other tested VOCs, including
acetone, acetaldehyde and 1-butanol, did not show interferences on carbon dioxide measured
by the PAS. This is likely because no overlap of the gas infrared adsorption spectra exists
between these VOCs and carbon dioxide. As expected, methanol in nitrogen also caused
interference on carbon dioxide (CF =0.35) slightly lower than methanol in air.
Besides, $SF_6$ measurements were interfered by the tested non-targeted VOC, with lower
correction factor obtained compared to $NH_3$, $CH_4$, $N_2O$ and $CO_2$. Acetic acid and 2-butanone
caused the highest interferences on $SF_6$, with correction factors of 0.31 and 0.23, respectively.
Other tested VOCs caused significantly less interference on $SF_6$, among which methanol gave
the highest negative correction factor of -0.15. Again, the methanol in nitrogen gave a
significantly lower level of interference on $SF_6$ compared to methanol in air (CF = -0.056 vs -

444    0.15).

Overall, the tested non-target VOCs in this study caused significant interference on target gases,
where ammonia and methane showed the most interference. Even though less interference was
observed for nitrous oxide, this could still cause problems due to the typically low concentration
level of this compound in e.g. livestock facilities or soil (Iqbal et al., 2013; Rong et al., 2014).
Nevertheless, the correction factors obtained from this study offer a possibility for correcting





for the interferences caused by the tested non-targeted VOCs, if the specific VOC
concentrations are available from simultaneous measurements. For historical data this is apart
from a few exceptions never the case.



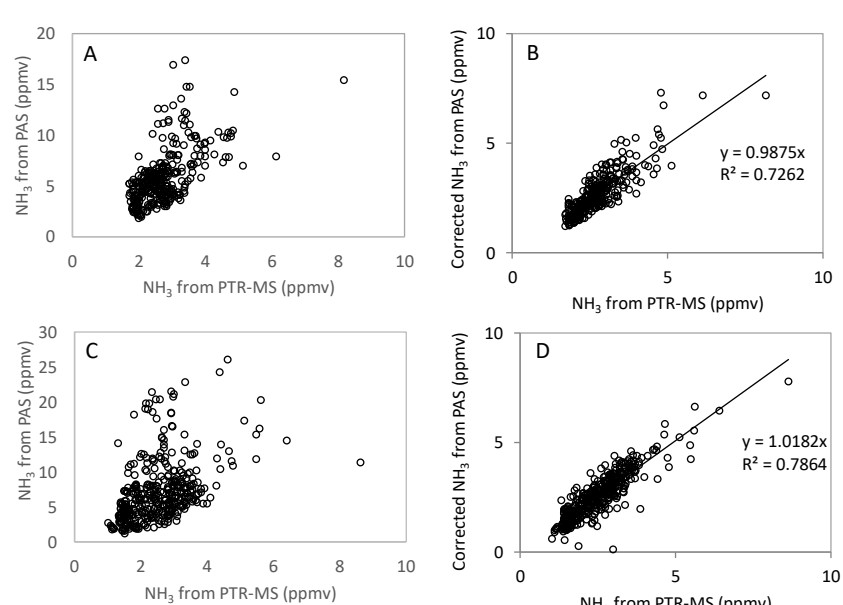

**Figure 5.** $NH_3$ concentrations measured by the PAS (vertical axis) and by the PTR-MS (horizontal
axis) in the field measurement from Location One before the correction by the tested non-targeted
VOCs (A) and after the correction by the tested non-targeted VOCs (B), and from Location Two
before the correction by the tested non-targeted VOCs (C) and after the correction by the tested non-
targeted VOCs (D).

**3.4     Experiment 4: Field test for validation of correction factors**
During the field test in the partially ventilated dairy barn, the ammonia measurements by PAS
and PTR-MS were compared between each other for one location in the pit and two locations
(Location One and Location Two) in the barn. Figure S2 showed the ammonia measured by



PAS and PTR-MS at the measurement point of the pit ventilation. In the pit ventilation, low
concentrations of VOCs were generally obtained and relatively high concentrations of ammonia
were observed for both instruments. Thus, no significant interferences were observed for
ammonia measured by the PAS, and ammonia measurements by PAS and PTR-MS showed a
good agreement as shown in Figure S2. However, for the two measurement points inside the
barn, significantly higher ammonia concentrations were obtained from PAS compared to the
concentrations measured by PTR-MS (Figure 5 A & C). The higher ammonia concentration
observed for the PAS measurement was most likely due to the interferences from VOCs, some
of which had high concentrations, especially for ethanol as shown in Table 3. In fact, the relation
between the ammonia concentrations measured by PAS and the ethanol concentrations
measured by PTR-MS, were highly correlated for both measurement locations, with slopes
close to 3 (2.97 and 3.12; see Figure S3). These two numbers are generally close to the
correction factor obtained for ethanol (CF = 2.81). The correction factors obtained in
'Experiment 3' were used for data correction of ammonia measurement by PAS since the
instrument configurations were kept the same. Thus, the interference of the VOCs on ammonia
measurement by PAS could be estimated from the correction factors obtained in 'Experiment
3' and used to correct the ammonia data. Figure 5B & D show the corrected ammonia
concentrations measured by PAS by using the correction factors, together with the measured
ammonia concentration by the PTR-MS for both measurement locations. The corrected
ammonia concentrations from the PAS are generally in good agreement with the ammonia
concentration measured by the PTR-MS, with slopes were close to 1 (0.99 and 1.02). This
experiment validated that with the correction from major VOCs, the interference on NH3





measured by PAS could be reasonable estimated in field applications. However, it should be
noted that a lot of redundant work is needed to make this correction if only $NH_3$ concentration
determination needed, since a number of VOCs concentrations need to be known in order to
achieve a right correction even though some minor VOCs within low range ppbv could be
omitted.

**Table 3.** Average concentrations ($\pm$standard deviation) of selected VOCs during the field test in the
dairy cattle barn for the two sampling locations 1 & 2, both of which are located inside the barn.

| Compound | Concentrations (ppbv) | |
|---|---|---|
| | Location 1 | Location 2 |
| ethanol | 1421±946 | 1622±1355 |
| methanol | 237±150 | 241±192 |
| acetic acid | 57±41 | 69±62 |
| acetaldehyde | 99±81 | 92±84 |
| 2-butanone | 19±11 | 17±13 |
| acetone | 78±30 | 52±25 |
| 1-propanol | 71±45 | 72±68 |
| 1-butanol | 22±10 | 16±12 |
| hydrogen sulfide | 12±10 | 11±8 |
| trimethylamine | 8.6±3.5 | 5.7±3.1 |
| dimethyl sulfide | 15±9 | 14±10 |
| 4-methylphenol | 5.2±2.1 | 3.8±2.2 |



**4   Conclusions**
One must take special care when measuring $NH_3$ and greenhouse gas emissions by using PAS
techniques as Innova. Depending on the IR absorption spectra of different gases, non-targeted
gases such as VOCs may interfere significantly with the target gases causing inaccurate results.
In order to confirm and determine the correction factors regarding the interference on targeted
gases caused by selected VOCs, experiments were conducted by using simultaneously a PAS
and a PTR-MS, while also clarified by a CRDS. Results from these experiments provide useful



guidelines with regards to interferences caused by non-targeted gases. The results on correction
factors revealed that the tested VOCs of ethanol, methanol, 1-butanol, 1-propanol and acetic
acid caused the most significant interference on $NH_3$ measured by PAS. Interestingly, non-linear
relations were obtained for interreferences on $N_2O$ by test VOCs as non-targeted gases, while
linear response was obtained for interference on other targeted gases. The field test in the cattle
barn validated the interference caused by VOCs on $NH_3$ measurement by the PAS when
simultaneously measured by the PTR-MS. Therefore, the correction factors could be used for
potential data corrections when same type of PAS is used together with available VOCs data.
No validation was performed for greenhouse emissions correction due to lack of alternative
measurement.

*Code and data availability*. Data and code are available upon request to the corresponding
author.
*Supplement*. The supplementary information is available free of charge at DOI: .

*Author contributions*. DL, LR and AF designed the setup for the experiments performed; LR,
XK and AC contributed to setting up and conducting experiments and acquiring data; DL, AF,
JK, XK and APA contributed to section writing and analysis; LR, AF and JK assisted in data
analysis and manuscript editing.

*Competing interests*. The authors declare that they have no conflicts of interest.





*Acknowledgements*. This work was supported by National Natural Science Fund of China
(No. 31672468) and Thousand Talents Program (Youth Project 2016).

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
