# Peer review of "Photoacoustic measurement may significantly overestimate NH₃ emissions from cattle houses due to VOC interferences"

_Atmospheric Measurement Techniques, 2018_

## Referee Comment (RC2)

Review of the manuscript "Photoacoustic measurement may significantly overestimate NH3 emissions from cattle houses due to VOC interferences" by Liu et al. (amt-2018-412)

General comments

The paper investigates interferences caused by volatile organic compounds (VOCs) in the photoacoustic measurement of ammonia in air on agricultural farms. The commercially available PAS instrument that was used in this work contains a broadband infrared light source and six optical filters for spectral selection. The filter set is suitable for the specific detection of NH3, CH4, CO2, H2O, N2O and SF6 in atmospheric mixtures at ppmv level, if no other absorbing species interfere. The experiments described in the manuscipt show very clearly that VOCs, such as methanol, ethanol or acetic acid observed in a dairy farm can cause significant interferences and are detected by the PAS instrument with higher sensitivity than ammonia. Similar results are found for greenhouse gases that are measured by the PAS instrument. A comparison to a reference technique (PTR-MS), which measured ammonia and VOCs, demonstrates that the PAS instrument can overestimate ammonia concentrations by up to an order of magnitude due to spectral interferences by VOCs. The authors conclude that the PAS measurements of ammonia can be corrected, if the concentrations of the interfering VOCs are known.

This work is relevant as the described type of PAS instrument has been widely used for emission measurements of ammonia and greenhouse gases in agriculture. Although there have been experimental indications of interferences in the literature, there is little quantitative information available about cross-sensitivities to trace gases such as VOCs that can be co-emitted on animal farms. The paper presents new results, which help to understand the quality of PAS measurements of agricultural emissions. As such emissions play a role for air quality and climate, the topic of the paper is suitable for AMT. However, before it can be accepted for publication, major revisions of the manuscript are needed as outlined in the following.

1. The discussion about the general applicability of the PAS instrument for measurement of ammonia and greenhouse gases in agriculture needs to be broadened.

It should be pointed out more clearly that the technique used in the Innova instrument "is capable of measuring almost any gas that absorbs infrared light" (quotation from the Lumasense web page). The document "Detection limits for various gases" on the Lumasense web page lists almost 300 different organic gases that can be detected at the ppmv level. The method is based on non-dispersive broadband spectroscopy and selectivity is achieved by using appropriate wavelength filter, with one filter for each targeted trace gas. If the number of absorbing gases is larger than the number of optical filters, possible interferences are no surprise! The key questions are: (a) what is the magnitude of interferences that can be expected in agricultural environments, and (b) is it possible to quantify and correct interferences in a reasonable way? Both questions are adressed by the authors, but a more detailed discussion is needed.

(a) The authors show in Figure 5A and 5C that the PAS measurements of ammonia are higher than by PTR-MS, but quantitative statements of the measurement errors and interferences are missing in the text. The figure shows discrepancies between the two techniques of up to a factor of ten. How much of the difference can be explained by calibration errors? What is the measurement precision of the data points in Figure 5?
If the PTR-MS measurements are assumed to be interference free (is this a justified assumption?),

the discrepancies must be due to interferences in the PAS measurements, which apparently can be up to an order of magnitude greater than the correct ammonia concentration. The overestimation should be explicitly quantified!

(b) The authors argue that the interferences can be corrected if the interfering VOCs are measured by another technique (line 504 - 508). I agree that this would require a considerable additional effort. This effort should be explained in more detail. How accurate should the corrected data be? What accuracy and precision does this mean for measuring VOCs and determining their correction factors? The considerable additional effort appears to offset the advantage of the easy handling of the PAS instrument. It should therefore be discussed if alternative measurement techniques, for example CRDS (available for NH3, CH4, CO, CO2, H2O) would offer a better performance than PAS for agriculture emission measurements.

The authors conclude that the interferences listed in Table 2 can be corrected. I am not completely convinced that this is generally true. In the example in Figure 5, the interference is dominated by a single VOC (ethanol). Would the correction also work if two or more VOCs caused a similarly large interference at the same time? Are the interference corrections of individual substances (Table 2) independent of each other? See also Comment 25 below.

2. The title of the paper is not sufficiently descriptive. Spectral selection in PAS can be achieved in different ways with different specificities, for example, by optical filters in combination with broadband light sources (this work), by narrow bandwith lasers, interferometry, etc.. The title should inform about the technique that was used in this work. E.g. "Photoacoustic measurement using a broadband light source and optical filters may significantly overestimate ...".

3. The paper needs language editing which will eventually be provided by the publisher during the production process.

Reference

https://www.lumasenseinc.com/FR/produits/gas-sensing/gas-monitoring-instruments/photoacoustic-spectroscopy-pas/photoacoustic-gas-monitor-innova-1512/photoacoustic-gas-monitor-innova-1512.html.

Specific comments

1. Line 24 and line 40. Specify the greenhouse gases to which you refer.
2. Line 28 - 37. Be more quantitative and specific! What were the concentration levels of ammonia and VOCs which were studied in this work? Was is meant by "various levels of interference"? Quantitative information of cross-sensitivities (ppmv of false ammonia/ppmv of VOC) should be given here. How large were the corrections (order of magnitude) which were applied to the PAS readings in the field experiments? What is meant by "reasonably correlated" (Line 35)?
3. Line 51 - 56. The concentration range in emission studies of ammonia and greenhouse gases in agriculture should be quantified and distinguished from concentrations in rural background air. What are the analytical requirements (concentration range, time resolution, limit-of-detection) for measurements in a dairy farm?
4. Line 55 - 56. Which methods were compared? What was the reason for the 30% discrepancies?

5. Line 62 ff. Here and in Section 2.1, more details of the measurement principle of PAS should be given. What kind of light source is used (spectral range, emission bandwith, continuous or pulsed)? What causes the cell pressure changes? How are they detected? What is the range of optical absorbances? Is the signal linearly dependent on the concentration of each analyte? Does the method need regular calibration by the user and how is calibration achieved?

6. Line 74 - 77. How is the cross compensation achieved? Is it performed online in the instrument or by postprocessing of raw data by the user? How large are typical corrections and what is the residual error?

7. Line 117 - 172. Please compare the specifications for ammonia measurements by PAS, CRDS, and PTR-MS in a table. It should include the lower limit of detection (LOD), dynamic range, response time, measurement time, accuracy, possible interferences.

8. Line 130. It is not clear, what was tested. Calibration procedure? PTR-MS instrument? How was the PTR-MS calibrated?

9. Line 148. It is not clear, what was tested. See Comment 8.

10. Line 162 - 171. See Comments 5 and 6.

11. Line 166. The filter for H2O is missing in Table S1. Are the filters for the other target species correctly named in Table S1? According to the document "Detection limits for various gases" (Lumasense web page*) filters "936" and "972" provide no sensitivity for ammonia and SF6, respectively. [*https://www.lumasenseinc.com/FR/produits/gas-sensing/gas-monitoring-instruments/photoacoustic-spectroscopy-pas/photoacoustic-gas-monitor-innova-1512/photoacoustic-gas-monitor-innova-1512.html]

12. Line 168 - 171. Have you checked the validity of the cross compensation of interferences from target gases or do you rely on the specificiations given by the supplier of the instrument?

13. Line 188 - 190. Is there a reason to assume that the time response of the PAS instrument depends on the magnitude of the ammonia concentration?

14. Line 191 - 207. The purpose of the experimental setup (Figure 1) is not entirely clear. Was is used as a quantitative source of VOC concentrations for calibration of the PTR-MS? If so, which method was applied to determine the VOC concentrations in the gas phase? Details should be given! Or was the setup simply used as a source of VOC/air mixtures and the VOC concentrations were measured by a calibrated PTR-MS. If this is case, how was the PTR-MS calibrated for the selected VOCs?

15. Line 206 - 207. The selected components (methanol, ethanol etc.) should be mentioned with reference to Section 3.2.

16. Line 212 - 220. What is the meaning of a "pre-tested water solution" (Line 213) ? What were the concentrations of the liquid solutions and their temperature? How were the VOC concentrations in the gas phase quantified? How stable were the gas-phase concentrations? How large was the water vapor concentration in the diluted gas which was fed into the instruments?

17. Line 221 - 224. What is the meaning and function of "excess flow". The excess flow should be indicated in Figure 1.

18. Line 236. Which company has manufactured the Multiplexer 1309? How does it work and what are the materials that come into contact with the sampled air?

19. Line 244. Which "selected VOCs" and "odorants" were measured? The term "odorant" should be defined.

20. Line 245 - 246. What is meant by "background"? Where were the four sampling lines going to?

21. Line 251 - 254. For which compounds were calibrations performed? The suppliers of the calibration gas mixtures and permeation device should be mentioned. What is the accuracy of the calibrations?

22. Line 281 - 282. "Background concentrations of ammonia measured..." can be misunderstood and should be rephrased. Do you mean instrumental baseline (instrumental offset) when ammonia-free zero air is measured? Is there a plausible explanation for the background values of the CRDS and PAS instruments?

23. Line 292 - 296, Figure 2B. The result of the instrumental comparison needs more discussion. Are the calibrations of the two instruments (CRDS, PTR-MS) independent? What is the statistical error of the slope of the linear fit? Are the differences between the two instruments statistically significant and can they be explained by instrumental calibration errors? The measurement comparison between PAS and CRDS (or PTR-MS) should also be shown and discussed.

24. Line 300 - 315. What could be the reason for the concentration dependence of the PAS response time? Why are the PAS values elevated at 18:43 (Fig. 2C) and decrease until 19:00, while CRDS and PTR-MS show constant values.
After the ammonia concentration has been switched to zero, the decay of the PAS signal seems to have at least two time constants. There is an initial fast decay that is followed by a tail with a slow decay. What could be the reason for the time behaviour? How long does it take until the PAS signal reaches baseline values?

25. Line 378 - 380. Interferences in ammonia PAS measurements have been studied for single VOCs in air. Have you tested whether the interferences are additive in a multicomponent gas mixture? Additivity would at least require that the optical absorbances of the interfering VOCs are small (avoiding line saturation). (Non)linearity of the relationships seen in Figure 4 may give useful information. For a mixture with multiple interfering VOCs, a valid correction using the information from Table 2 can only be expected, if the interferences are independent of each other. This aspect needs to be discussed.

26. Table 2. For which concentration ranges were the relationships tested? This information should be included in the table. Is there a physical explanation for negative interferences?

27. Line 412 - 425. Is there a physical explanation for the nonlinear behaviour of the VOC interference in measurements of N2O? Are the nonlinear interferences additive when two or more interfering VOCs are present in measured air?

28. In Figure 2B, 4, 5B, 5D, S2, S3 and Table 2, results from statistical data treatments are shown. More information should be given on how fitted lines (curves) and fitted parameters were obtained. Were measurements corrected for offsets (background)? How were linear fits obtained (least square fits? with or without error weighting?). Is it justified to force the fit through the origin? Fitted parameters should be given with $1\sigma$ errors. The meaning of the plotted error bars in Figure 4 should be explained. The error bars (statistical errors?) are significantly larger than the scatter of the data points around the fitted line. What does that mean?

Technical comments

1. Materials and methods. More information about used gases (nitrogen, zero air), chemicals (organic compounds listed in Table 2) and the water used for VOC solutions should be given.

Suppliers and purity grades need to be specified. How was zero air generated? How were the solutions of VOCs in water prepared?

2. Line 34. The term "field study" may be confused with a study under natural ambient conditions. Change sentence to "measured by PAS in a dairy farm"

3. Line 41 - 42. Ammonia causes soil acidification?

4. Line 67 - 70.  PAS is a general term for a spectroscopic method, but here you refer to a particular instrument (Innova 1312). Therefore, it is better to say: "Besides, the Innova 1312 has the advantages ...".

5. Line 69. Change to "Usually, water vapor is also measured in order ..."

6. Line 73. Change to "absorption of infrared light"

7. Line 80. There is a word missing in "interference of has not been well studied..."

8. Line 81. "Mathot et al., 2007" is missing in the Reference section.

9. Line 118-199: Change to "were used to measure trace gas concentrations in air".

10. Line 125 - 127. The sentence is not clear and should be rephrased.

11. Line 141. Either "acceleration of the ring down " or "reduction of ring down time ".

12. Line 143. "Normal" ring down time needs to be explained.

13. Line 173. Change to "Instrumental background signals, ammonia calibrations and instrumental time responses were characterized for the PAS, PTR-MS and CRDS instruments".

14. Line 180. Delete "for the calibration test".

15. Line 181. The desired concentration range should be specified.

16. Line 197. Which kind of plastic material was used for the container? Was it emission free?

17. Line 254. A reference for "Standard conditions as described previously ..." should be provided.

18. Line 293. Change to "in which the slope of the fitted line (k = 0.96 ± ?) ..."

19. Line 304. "90% decay time" needs to be defined.

20. Line 380. Change to "...VOC concentrations be measured simultaneously by other instruments".

21. Line 382. The term "correction factor" should be defined.

22. Figure 1. The exhaust line should be marked.

23. Figure 2. All axis should show tic marks. Concentrations at the x and y axis of Figure 2B and at the y axis of Figure 2C should be given in ppmv.
Figure 2C caption: change to "Instrumental response of PTR-MS, PAS and CRDS instruments to a rectangular ammonia concentration pulse."
Figure 2D caption: change to "Instrumental response of PTR-MS, PAS and CRDS instruments to a stepwise increase in ammonia concentration."

24. Figure 4. All axis should show tic marks. Concentrations at the x and y axis of Figure 4A and 4B should be given in ppmv. The Figure caption mentions red, green and purple lines. The colour designation should be made consistent with the plotted lines.

25. Figure 5. All axis should show tic marks. Concentrations at the x and y axis of Figure 5B and 5D should be given in ppmv. Draw 1:1 lines in A and C as reference.

26. Figure S1.  Concentrations should be given in ppmv.

27. Figure S2. All axis should show tic marks. "ppm" should be "ppmv".

28. Figure S3.  All axis should show tic marks. Concentrations should be given in ppmv. What is the difference between the upper and lower panel in Figure S3?

29. Table 1. Units should be ppmv (to be consistent with text and figures).

30. Table 2 caption must be rephrased. The table does not show correction factors, but empirical relationships describing the functional dependence of the interference in the measurement of the target compound (e.g., NH3) on VOC concentrations.
31. Table 3. Do standard deviations apply to the mean values, or do they represent the variability of single measurements? The concentration values for ethanol and methanol should be reasonably rounded.
32. Table S1. "Filter center" should be changed to "Center wavelength". What is the meaning of "Band width (%)" ? Is it related to "Filter bandpass"?

---

## Referee Comment (RC1) · Anonymous Referee #2 · 19 Feb 2019

**Photoacoustic measurement may significantly overestimate $NH_3$ emissions from cattle houses due to VOC interferences**

Dezhao Liu[1,2*], Li Rong[2], Jesper Kamp[2], Xianwang Kong[1], Anders Peter Adamsen[3], Albarune Chowdhury[2], Anders Feilberg[2*]

**General comments**

This manuscript describes experiments investigating VOC interference during NH3 measurements using PAS in cattle barn emissions.

The novelty is the analysis of VOC interference. PTR-MS measurmenst were used for this purpose. The experimental design is sound. Some details of M&M should be worked out.

Correction factors are proposed which should improve emission factor determination for cattle barns emissions. The necessity to perform both PTR-MS as well as PAS complicates the emission measurements.

In the discussion attention could be paid to the pissble transferability to other emission sources e.g. pig farm emissions.

**Detrailed comments**

190. her CRDS is introduced without any further explanation.  In line 96 "two PAS instruments …" are mentioned. Ths is confusing. Please  explain why CRDS is included.

195-199. The hedaspace in the silage box was analused by PTR-MS tehrefore a floxw of 75 ml/min was withdrawn. Was here any balance gas introduced into the bow compensating the sampled air?

203. What is meant by "pretested"?

218. More information should be given on how the water solution containing VOC was prepared.  pH? Concentrations?  …

Table 1. Suggestion: use consequently (throughout the paper) the acronyms for the methods (PTR-MS; CDRS; PAS) not the instrumental brand names.

Figure 3. How were the concentrations calculated? Just by the instrumental data base or own calibration.

Figure 4 Why ppbv in [A] and [B] and ppmv in [C] and [D]? Are the concentration ranges (acetic acid up to 40 ppmv) realistic for cattle barn emissions?

Table 2. For some compounds the correlations are based on 4 data points only (N). Validity?

394-396. What is the meaning for field measurements?

Table 5. Explain in the M&M section how the concentrations of the individual compounds were calculated.

Table 5. Acetic acid concentrations here are between 50 and 100 ppbv. In Figure 4 concentrations up to 40 ppmv are tested. Relevance?

539. Which greenhouse gases. Specify.

---

## Author Comment (AC1) · 23 Mar 2019

Response to Interactive discussion: 'amt-2018-412', Anonymous Referee #2, 19 Feb 2019:

General comments This manuscript describes experiments investigating VOC interference during NH3 measurements using PAS in cattle barn emissions. The novelty is the analysis of VOC interference. PTR-MS measurmenst were used for this purpose. The experimental design is sound. Some details of M&M should be worked out. Correction factors are proposed which should improve emission factor determination for cattle barns emissions. The necessity to perform both PTR-MS as well as PAS complicates the emission measurements. In the discussion attention could be paid to the pissble transferability to other emission sources e.g. pig farm emissions.

Response: Thank you for the comments. Some details of M&M are now clarified, according to the detailed comments from the referee. We do also agree that the measurements performed both by PTR-MS and the PAS complicates the emission measurements, but also give more detailed information and confirmation regarding measured emissions from cattle barn. Usually the VOCs from cattle barns are significantly lower in pig farm compared to cattle barn, so the interreference should be lower. However, the transferability should certainly be careful since the source the level of the VOCs pollutants are significantly different, as suggested by the referee.

Detrailed comments 190. her CRDS is introduced without any further explanation. In line 96 "two PAS instruments . . ." are mentioned. Ths is confusing. Please explain why CRDS is included. Response: Thanks for the comments. CRDS was first introduced in Line 109, and the technique was explained at Line 141 to Line 161. In line 96, "two PAS instruments . . ." means that the reference of Hassouna et al. (2013) used two PAS instruments for their study. This is now clarified in the manuscript and please see Line 96. The reason why CRDS was included was to further confirm that no ammonia was presented while VOCs was measured by the PTR-MS and caused interference when measured by PAS. Due to the fact that the PTR-MS had high background on measuring ammonia, the measurement by the CRDS meanwhile could therefore make a solid and second confirmation of our measurement.

195-199. The hedaspace in the silage box was analused by PTR-MS tehrefore a floxw of 75 ml/min was withdrawn. Was here any balance gas introduced into the bow compensating the sampled air? Response: Thanks for the comment. Indeed 75 ml/min was withdrawn from the headspace in the silage box, while another 75 ml/min of zero air was supplied and controlled by a mass flow controller, before measured by the PTR-MS (inlet flow was set to 150 ml/min). Since the silage box was not closed, with two oval holding holes on sides, therefore the balance gas from the room air would compensate the sampled air from the headspace in the silage box. 203. What is meant by "pretested"? Response: Thanks for the comment. The pre-test for water solution preparation used a ratio of VOC:Water as 1:5, and the ratio between VOC and water was adjusted if the purged concentration after dilution (by zero air controlled by 2 mass flow controllers) measured by the PTR-MS was not within the desired range (too low or too high). Please see the explaination at Line 221-224.

218. More information should be given on how the water solution containing VOC was prepared. pH? Concentrations? ... Response: Thanks for the comment. The pre-test for water solution preparation used a ratio of VOC:Water as 1:5, and the ratio between VOC and water was adjusted if the purged concentration after dilution (by zero air controlled by 2 mass flow controllers) measured by the PTR-MS was not within the desired range (too low or too high). The pH was not measured for the water solution. The concentration level was varied depending on flow rate and an example could be seen in Figure 4.

Table 1. Suggestion: use consequently (throughout the paper) the acronyms for the methods (PTR-MS; CDRS; PAS) not the instrumental brand names. Response: Thanks for the suggestion. The instrumental brand names are now avoided as far as possible throughout the paper, and the acronyms for the methods (PTR-MS; CDRS; PAS) are used instead.

Figure 3. How were the concentrations calculated? Just by the instrumental data base or own calibration. Response: The PTR-MS can measure VOC concentrations directly, and the calculation of VOC concentrations by the PTR-MS was depended on a number of parameters especially the reaction rate between VOC and protonated water. Nevertheless, the ethanol was corrected and calibrated separately according to ethanol fragmentation.

Figure 4 Why ppbv in [A] and [B] and ppmv in [C] and [D]? Are the concentration ranges (acetic acid up to 40 ppmv) realistic for cattle barn emissions? Response: Thanks for the comment. All the concentration unit in Figure 4 are now revised to ppbv. Acetic acid of 40 ppmv might not be common for cattle barn emissions, but could happen within a short period at a specific location such as nearby the silage and within feeding.

Table 2. For some compounds the correlations are based on 4 data points only (N). Validity? Response: Thanks for the comment. Indeed, in some cases the data points were only 4, which might need further validation in such cases. Still, the correlation coefficients showed in the table seems to be reasonable, indicating the validity of this method and data.

394-396. What is the meaning for field measurements? Response: The field measurements means the measurements performed in the field for a full-scale cattle barn.

Table 5. Explain in the M&M section how the concentrations of the individual compounds were calculated. Response: Thanks for the comment. I guess the referee means "Table 3" here, where the VOC concentrations were determined directly by the PTR-MS, based on estimated reaction rate constants described by Liu et al. (Liu et al., 2018). The manuscript now is revised accordingly and please see Line 259-260.

Table 5. Acetic acid concentrations here are between 50 and 100 ppbv. In Figure 4 concentrations up to 40 ppmv are tested. Relevance? Response: Thanks for the comment. The averaged acetic acid concentration for Location 2 was 69 ppbv with standard deviation of 62 ppbv. Since the measurement location was not close to the silage feeding in the cattle barn, the concentration of acetic acid near the location of silage feeding might be significantly higher than this concentration range. Another reason for performing high acetic acid concentration up to 40 ppmv was to investigate the linear range of the correction factors found in this study.

539. Which greenhouse gases. Specify. Response: Thanks for the comment. The greenhouse gases are CH4, N2O and CO2, which is now specified in the revised manuscript. Please see Line 518-519 in the revised manuscript.

Please also note the supplement to this comment:
https://www.atmos-meas-tech-discuss.net/amt-2018-412/amt-2018-412-AC1-
supplement.pdf

**Supplement:**

[revised manuscript text omitted]

---

## Short Comment (SC1) · 4 Apr 2019

It has been reported that the measurements of ammonia around livestock facilities by PAS could be interfered by the VOCs, a systematic calibration is still in great need. This research conducted precisely test and calibration on the interference of VOCs on the measurement of NH3, NO2, CH4 by PAS method. The results confirmed the significant influence of the VOCs on the measurement of these compounds. Moreover, the authors dedicate to building correction factors for the interference and verifying the factors in the field tests successfully. Overall, this is high-quality research and the results deserve to be carefully read by researches on field measurement of NH3, NO2,

CH4 by PAS method.

---

## Short Comment (SC2) · 25 Jun 2019

We were always surprised how confident researchers have been in trace gas instruments using broadband light sources for the determination of gaseous emissions from agriculture. The present paper focuses on interferences occurring in ammonia concentration determination by INNOVA instruments. It convincingly shows the severe limitations of this measurement technology and documents the strong influence of other gases on the readout of these devices. Reliable concentration measurements need in addition the determination of a large variety of compounds in order to perform the suggested correction algorithms. This is a paradoxical situation, as such measurements

need complex and costly instrumentation that will make the use of broadband filter-based instruments redundant. In some cases cheaper NH3 passive samplers or liquid impingers can be used as a robust and reliable alternative, if high time resolution is not required. Innova-based NH3 concentration measurements are likely influenced by compensating errors. The positive interferences discussed in this paper focus on the detection side, but in parallel there are potential losses of NH3 in the inlet tubing/lines and switching valves, especially in multiport systems. Within livestock production systems rather high dew points are present that facilitate absorption onto most types of surfaces. Often low flow rates are used, tubing is not heated, and some inlet filters are put in place to protect the analyzer from dust ingress, which all exacerbate the adsorption/desorption problem.

It is also striking that in many peer-reviewed articles reporting emission data based on Innova or similar instruments, detailed information on the sampling and analytical system are missing, which prevents a critical re-evaluation of such data. Consequently, emission factors based on these instruments should be taken with great caution.

Fifteen years ago, we investigated in detail the dependence of the concentration output, for a large range of CO2-CH4-N2O-H2O mixtures, of three different Innova's (1312, 1314) as a function of the cell temperature and the water vapor (dew point). An overview of the measurements is given in the supplement. These analyses are the base of the evaluation algorithm developed by Flechard et al. (2005) for the determination of N2O concentrations, which showed the large influence of CO2, water vapour and temperature. Another striking result was the large influence of the water content and the cell temperature on the CH4 concentration. The contribution of H2O to the CH4 raw signal depends on the ratio of the two gases. It will e.g. be less important in cow stable, compared to our INNOVA experiment where CH4 was only a couple of ppm.

Only in case Innova devices are operated in a temperature-controlled environment, with the dew point being kept constant, can reliable CH4 measurements be made.

Reference Flechard, C. R., Neftel, A., Jocher, M., Ammann, C., and Fuhrer, J.: Bidirectional soil-atmosphere N2O exchange over two mown grassland systems with contrasting management practices, Glob. Change Biol., 11, 2114–2127, 2005.

Albrecht Neftel Neftel Research Expertise Christoph Flechard INRA, Rennes

Please also note the supplement to this comment:
https://www.atmos-meas-tech-discuss.net/amt-2018-412/amt-2018-412-SC2-supplement.pdf
* * *
[Figure]

**Supplement:**

**Performance of three INNOVA analyzers for ambient Greenhouse Gas Measurements**

Albrecht Neftel
Neftel Research Expertise, Oberwohlenstrasse 27, CH-3033 Wohlen b.Bern, Switzerland
Christoph Flechard
1INRA, Agrocampus Ouest, UMR1069 Sol Agro-hydrosyst`eme Spatialisation, 35042 Rennes, France

Photoacoustic (PAS) trace gas analyzers are based on the principle of light absorbance in a chopped beam that converts the absorbed energy into an acoustic signal that is recorded with a microphone. This technique is reliable and sensitive, but the selectivity depends mainly on the used light source and on the geometry of the absorption cell. E.g. the INNOVA family uses a broadband light source with a selection of suitable wavelength by interference filters.

[Figure]

Figure 1: Broadband filters used within the Innova family. As an example, the $N_2O$ transparency window includes absorption bands for $CO_2$ and $H_2O$.

In case the absorptions of all trace species in a given window are independent of each other and are strictly proportional to their molecular density (i.e. the amount of absorbing molecules in the light beam) the evaluation of the concentrations needs as many different filters as gases and correspond to the solution of a linear equation systems.

Reality is more complex. Effective absorption coefficients are concentration dependent and are influenced by other absorbing molecules in the window. In addition, the sensitivity of the detection strongly depends on the cell temperature and also on the water concentration.

In 2004 we tested the performance of three INNOVA instruments to measure CO2, N2O, CH4 and H2O in concentration ranges as they typically appear in ambient air in ecosystem research (which, incidentally, are much lower than in the confined air of animal houses). Using the interference correction certified by the manufacturer, the data showed strong deviation for $N_2O$ and $CH_4$ that systematically depended on water vapor and cell temperature.

Figure 2 shows the effect of varying CO2 and H2O concentrations on the raw signal strength of the N2O filter UA0985 for a $N_2O$ mixing ratio of 300 ppb, ie of the order of the mean atmospheric level . The figure shows that the raw signal varies over one order of magnitude and strongly depends on the cell temperature and water vapour concentration.

**N2O raw photoacoustic signal 300 ppb : interference by H2O and CO2**

**(INNOVA 1312 - N2O filter UA0985)**

[Figure]

Figure 2 shows the effect of varying CO2 and H2O concentrations on the raw signal strength of the N2O filter UA0985

Figures 3a-c shows the comparison of given standard concentrations and the Innova measurements using the instrument's default correction algorithm for CO2, N2O and CH4, as function of varying water vapor level and cell temperature.
The data illustrate the strong and systematic deviations from the given concentration, that depends in a complex way from the cell temperature, the water vapor and the level of the other gases. How to read these figures? The experiments are always grouped by four values with fixed CO2, N2O and CH4 concentrations, but increasing cell temperature from 20 to 50°C. These groups demonstrate a very strong influence of the cell temperature. Between the first three groups the N2O concentration changed from 500ppb to 2500ppb and then to 5000ppb. The effect of the N2O concentrations e.g. on the CH4 concentration can then be seen by comparing e.g. always the last point of an individual group over the three groups. Most striking are the strong deviations in the calculated CH4 concentrations that even change the sign and are several times the effective concentration.
Nevertheless, the systematic pattern of the deviations seems to allow to develop a correction algorithm as it was done by Flechard et al. (2005)., but only up to a point: if the ratio of the interfering gas (eg $CO_2$ or $H_2O$) to the gas of interest (eg $N_2O$ or $CH_4$) was too

large, then a very large fraction of the raw signal given by the filter for the gas of interest is actually due to the interfering gas, and the interference can no longer reliably be corrected for due to the large noise in the data.

[Figure]

Figure 3a CO2 concentration determined by the Innova's instruments. The number on the x-axis refers to trials with conditions as given in the first panel. The solid lines in the panels indicates the applied standard gas concentration for CO2.

[Figure]

Figure 3b N2O concentration determined by the Innova's instruments. The number on the x-axis refers to trials with conditions as given in the first panel. The solid lines in the panels indicates the applied standard gas concentration for N2O

[Figure]

Figure 3c CH4 concentration determined by the Innova's instruments. The number on the x-axis refers to trials with conditions as given in the first panel. The solid lines in the panels indicates the applied standard gas concentration for CH4.

---

## Author Comment (AC2) · 27 Sep 2019

General comments

The paper investigates interferences caused by volatile organic compounds (VOCs) in the photoacoustic measurement of ammonia in air on agricultural farms. The commercially available PAS instrument that was used in this work contains a broadband infrared light source and six optical filters for spectral selection. The filter set is suitable for the specific detection of NH3, CH4, CO2, H2O, N2O and SF6 in atmospheric mixtures at ppmv level, if no other absorbing species interfere. The experiments described in the manuscipt show very clearly that VOCs, such as methanol, ethanol or acetic acid observed in a dairy farm can cause significant interferences and are detected by the PAS instrument with higher sensitivity than ammonia. Similar results are found for greenhouse gases that are measured by the PAS instrument. A comparison to a reference technique (PTR-MS), which measured ammonia and VOCs, demonstrates that the PAS instrument can overestimate ammonia concentrations by up to an order of magnitude due to spectral interferences by VOCs. The authors conclude that the PAS measurements of ammonia can be corrected, if the concentrations of the interfering VOCs are known.

This work is relevant as the described type of PAS instrument has been widely used for emission measurements of ammonia and greenhouse gases in agriculture. Although there have been experimental indications of interferences in the literature, there is little quantitative information available about cross-sensitivities to trace gases such as VOCs that can be co-emitted on animal farms. The paper presents new results, which help to understand the quality of PAS measurements of agricultural emissions. As such emissions play a role for air quality and climate, the topic of the paper is suitable for AMT. However, before it can be accepted for publication, major revisions of the manuscript are needed as outlined in the following.

Response: Thank you very much for the general comments and all valuable comments.

1. The discussion about the general applicability of the PAS instrument for measurement of ammonia and greenhouse gases in agriculture needs to be broadened.

It should be pointed out more clearly that the technique used in the Innova instrument "is capable of measuring almost any gas that absorbs infrared light" (quotation from the Lumasense web page). The document "Detection limits for various gases" on the Lumasense web page lists almost 300 different organic gases that can be detected at the ppmv level. The method is based on nondispersive broadband spectroscopy and selectivity is achieved by using appropriate wavelength filter, with one filter for each targeted trace gas. If the number of absorbing gases

is larger than the number of optical filters, possible interferences are no surprise! The key questions are: (a) what is the magnitude of interferences that can be expected in agricultural environments, and (b) is it possible to quantify and correct interferences in a reasonable way? Both questions are adressed by the authors, but a more detailed discussion is needed.

Response: In principle, the technique used in the Innova instrument "is capable of measuring almost any gas that absorbs infrared light". Now this is pointed out clearly in the introduction (Line 68-70). The two key questions "(a) what is the magnitude of interferences that can be expected in agricultural environments, and (b) is it possible to quantify and correct interferences in a reasonable way?" are now included in the introduction in order to help to clarify the purpose of this study. Please see Line 84-86 in the revised manuscript.

(a) The authors show in Figure 5A and 5C that the PAS measurements of ammonia are higher than by PTR-MS, but quantitative statements of the measurement errors and interferences are missing in the text. The figure shows discrepancies between the two techniques of up to a factor of ten. How much of the difference can be explained by calibration errors? What is the measurement precision of the data points in Figure 5? If the PTR-MS measurements are assumed to be interference free (is this a justified assumption?), the discrepancies must be due to interferences in the PAS measurements, which apparently can be up to an order of magnitude greater than the correct ammonia concentration. The overestimation should be explicitly quantified!

Response: In Figure 5A and 5C, the factor is typically between 1 and 5, with few cases close to 10 (see Table S3). Only a small part of this difference can be explained by calibration errors. For the PTR-MS, the calibration error is around 10-15%. For the Innova 1312, the calibration error is around 20%. The measurement precision (ratio of standard deviation/averaged concentration) of the data points in Figure 5 was around 1%-3%. For ammonia measurement by PTR-MS, the measurements are interference free. Thus the discrepancies are surely caused by the PAS measurements, with some overestimation as the reviewer mentioned (the corrected ammonia fit generally ok though, as showed in Fig.5B and 5D). The overestimation possibly due to the interaction effects by various VOCs on the interference correction, but the quantification is difficult since we did not investigate the effects of interference when simultaneous present of multiple VOCs.

(b) The authors argue that the interferences can be corrected if the interfering VOCs are measured by another technique (line 504 - 508). I agree that this would require a considerable additional effort. This effort should be explained in more detail. How accurate should the corrected data be? What accuracy and precision does this mean for measuring VOCs and determining their correction factors? The considerable additional effort appears to offset the

advantage of the easy handling of the PAS instrument. It should therefore be discussed if alternative measurement techniques, for example CRDS (available for NH3, CH4, CO, CO2, H2O) would offer a better performance than PAS for agriculture emission measurements.

The authors conclude that the interferences listed in Table 2 can be corrected. I am not completely convinced that this is generally true. In the example in Figure 5, the interference is dominated by a single VOC (ethanol). Would the correction also work if two or more VOCs caused a similarly large interference at the same time? Are the interference corrections of individual substances (Table 2) independent of each other? See also Comment 25 below.

Response: Indeed, if the VOCs could be measured by another technique such as PTR-MS, interferences could possibly be corrected, but considerable additional efforts needed for correcting the data, and obviously it does not make sense to buy a PTR-MS in order to correct a PAS. The accuracy of the corrected data should certainly be as higher as better and at least to be able to reflect the right concentration within lower range of ppbv. Thus the required accuracy and precision for the measured VOCs should also be relatively high as lower range of ppbv. This additional effort certainly offset the advantage of the easy handling of the PAS instrument, and alternative measurement techniques, for example CRDS may indeed offer a better performance than PAS for agriculture emission measurements. The recent publication by Kamp et al., 2019 demonstrated that CRDS (Picarro G2103, only measuring ammonia and water) can measure ammonia with great precision without interferences and therefore have a better performance compared to PAS.

Regarding the correction factors (now changed to empirical relationships as suggested) listed in Table 2, we have to acknowledge that we don't know if the effects are always additive – although they are likely to be. In the field study, ethanol certainly dominated the VOC matrix in general, but other types of VOC also contribute significantly. For example, the averaged ratio of ethanol concentration to the sum of the 8 VOCs (tested in lab with obtained correction factors) was 0.64 ($\pm$0.11) for Location Two in the field study. From this single application it seemed that the obtained empirical relationships to be additive, but a complete investigation is indeed needed in the future. This part is now revised accordingly and please see Line 552-560.

2. The title of the paper is not sufficiently descriptive. Spectral selection in PAS can be achieved in different ways with different specificities, for example, by optical filters in combination with broadband light sources (this work), by narrow bandwith lasers, interferometry, etc.. The title should inform about the technique that was used in this work. E.g. "Photoacoustic measurement using a broadband light source and optical filters may significantly overestimate ...".

Response: We agree with you, that we only tested PAS using a broadband light source and optical filters. We now have revised the title as suggested.

3. The paper needs language editing which will eventually be provided by the publisher during

the production process.

Reference

https://www.lumasenseinc.com/FR/produits/gas-sensing/gas-monitoringinstruments/photoacoustic-spectroscopy-pas/photoacoustic-gas-monitor-innova-1512/photoacoustic-gas-monitor-innova-1512.html.

Specific comments

1. Line 24 and line 40. Specify the greenhouse gases to which you refer.

Response: The greenhouse gases are now specified in the revised manuscript.

2. Line 28 - 37. Be more quantitative and specific! What were the concentration levels of ammonia and VOCs which were studied in this work? Was is meant by "various levels of interference"? Quantitative information of cross-sensitivities (ppmv of false ammonia/ppmv of VOC) should be given here. How large were the corrections (order of magnitude) which were applied to the PAS readings in the field experiments? What is meant by "reasonably correlated" (Line 35)?

Response: Yes, "various levels of interference" was meant for various VOCs levels which were tested for interference test while no ammonia was presented. We now add quantitative correction factors for highest VOC on ammonia interference (see Line 32-33). In the field experiment, data from corrected from PAS were correlated with the data from PTR-MS for ammonia, by applying the correction factors obtained from the single VOC test for interference. In the field experiments, the corrections for the PAS readings for Location One and Location Two were 2.14 (±0.75) and 2.88 (±1.85), respectively. The "reasonable" has been deleted in the sentence. These have been revised in the revised manuscript and please see Line 32-38.

3. Line 51 - 56. The concentration range in emission studies of ammonia and greenhouse gases in agriculture should be quantified and distinguished from concentrations in rural background air. What are the analytical requirements (concentration range, time resolution, limit-of-detection) for measurements in a dairy farm?

Response: The quantification of concentration range for ammonia and greenhouse gases in agriculture were done by a number of relevant studies previously. For example, Rong et al. (2015) quantified a dairy farm indoor and outdoor ammonia and greenhouse gases concentrations for both summer and winter periods, as follows: ammonia (indoor: 0.38-15.5 ppmv; outdoor: 0.2-3.3 ppmv); methane (indoor: 2.1-219 ppmv; outdoor: 1.1-11.1 ppmv); nitrous oxide (indoor: 0.19-0.83 ppmv; outdoor: 0.23-0.43 ppmv); carbon dioxide (indoor: 418-

2727 ppmv; outdoor: 402-646 ppmv). Although a lot of these quantification were performed by using PAS and the interference quantification were missing, these data could give some hints regarding the analytical requirements for the measurements in a dairy farm. For example, for nitrous oxide measurement in a dairy farm, the concentration range is between 200-300 ppbv and around 1 ppmv, time resolution should be as high as per few minutes in order to catch dynamic change, and the limit of detection needs a lower range of ppbv since the concentration level in a dairy farm is generally close to background air. PAS can meet this requirement but likely face interference problem.

4. Line 55 - 56. Which methods were compared? What was the reason for the 30% discrepancies?

Response: In the reference, three measurement techniques were used for measuring ammonia emission rates, e.g., external tracer ratio method (SF6 was used), internal tracer ratio (ITR) method (SF6 was also used), and flux sampler method. As the authors claimed, all three methods were validated, with however statistically significant biases for the measured release rate and no clear explanation for the biases was provided.

5. Line 62 ff. Here and in Section 2.1, more details of the measurement principle of PAS should be given. What kind of light source is used (spectral range, emission bandwith, continuous or pulsed)? What causes the cell pressure changes? How are they detected? What is the range of optical absorbances? Is the signal linearly dependent on the concentration of each analyte? Does the method need regular calibration by the user and how is calibration achieved?

Response: The infra-red light source is used. Innova 1312 used the filter of UA0982 (CO2), UA0985 (N2O), UA0936 (NH3), UA0972 (Freon 134a), and UA0969 (CH4). Infrared radiation can interact with a molecule and transfer energy to it if the frequency of the radiation is exactly the same as the frequency of a vibration within the molecule. When the molecule absorbs this radiation it vibrates with greater amplitude. This increased activity is short-lived, however, and the excited molecule very quickly transfers its extra energy to other molecules in the vicinity by colliding with them and causing them to travel more quickly. The increased molecular speeds means that the temperature in the measurement chamber increases and when the chamber is sealed the pressure also increases. The amount of light absorbed can be measured by measuring either the heat energy released or associated pressure increase. Both parameters are proportional to the concentration of the absorbing species. Because calorimetric detectors have slow response times and are insufficiently sensitive, the pressure increase is the preferred

measurement parameter. A microphone is an excellent detector of fluctuating pressure.
It is stated on the brochure of Innova 1312 'Linear response over a wide dynamic range'. On the
manual of Innova 1312, there is a chapter to explain how the users can conduct a self-calibration
after moving the instrument around. In our experiments, the instrument was calibrated by the

**Wavenumber/wavelength and bandwidth**

[Figure]

Fig. 1 Centre wavelength and half-power bandwidths of the optical filters

company and the instrument was calibrated on a certified gas cylinder of $NH_3$ in the lab. Now
more details are given in the revised manuscript, section 2.1 (please see Line 171-186).

6. Line 74 - 77. How is the cross compensation achieved? Is it performed online in the
instrument or by postprocessing of raw data by the user? How large are typical corrections and
what is the residual error?

Response:If the gas is targeted, the cross compensation can be achieved by the instrument
online. The user can also output the raw data. For example, water vapor absorbs infra-red light
at most wavelengths so that it will contribute to the total acoustic signal in the analysis cell no
matter which optical filter is used. A special optical filter is permanently installed in the filter
carousel of the 1312, which allows water vapor's contribution to the measured separately during
each measurement cycle. The 1312 is thus able to compensate for water-vapor's interference.
The corrections will be depending on the overlaps of the gas IR spectra but the residual error
after the cross compensation is negligible according to Zhao et al. (2012).

7. Line 117 - 172. Please compare the specifications for ammonia measurements by PAS,
CRDS, and PTR-MS in a table. It should include the lower limit of detection (LOD), dynamic

range, response time, measurement time, accuracy, possible interferences.

Response: These specifications are related to "experiment 1: laboratory test on ammonia calibration", therefore we now add relevant information to Table 1, as shown in the revised manuscript.

8. Line 130. It is not clear, what was tested. Calibration procedure? PTR-MS instrument? How was the PTR-MS calibrated?

Response: The ammonia calibration test was done by applying PTR-MS instruments, as described in the section "**2.2   Experiment 1: laboratory test on ammonia calibration**". The instrument calibration was performed based on specific reaction rate constants and transmission (accuracy better than 12%), as described in our previous study (Liu et al., 2018). This is now clarified in the revised manuscript at Line 144-146.

9. Line 148. It is not clear, what was tested. See Comment 8.

Response: The calibration test was done by applying the CRDS instrument (Picarro), as described in the section "**2.2   Experiment 1: laboratory test on ammonia calibration**". The accuracy of the CRDS instrument is routinely checked against a certified reference gas as described by Kamp et al (2019). This is now clarified in the revised manuscript at Line 158-160.

10. Line 162 - 171. See Comments 5 and 6.

Response: This is now revised as suggested. Pease see Line 171-178.

11. Line 166. The filter for H2O is missing in Table S1. Are the filters for the other target species correctly named in Table S1? According to the document "Detection limits for various gases" (Lumasense web page*) filters "936" and "972" provide no sensitivity for ammonia and SF6, respectively. [*https://www.lumasenseinc.com/FR/produits/gas-sensing/gas-monitoringinstruments/photoacoustic-spectroscopy-pas/photoacoustic-gas-monitor-innova-1512/photoacoustic-gas-monitor-innova-1512.html]

Response: The filter for $H_2O$ is always included so this filter typically was not considered as other filters which could be different from instrument to instrument. Thus we didn't include $H_2O$ filter in Table S1, while other target species are correctly named and included, with information provided by the producer. The detailed optical filters can also be seen from the response for comment 5. Since the version of 1512 is different than the version we used for the experiment, it is likely they updated the filters in the newer version of instrument.

12. Line 168 - 171. Have you checked the validity of the cross compensation of interferences from target gases or do you rely on the specificiations given by the supplier of the instrument?

Response: We did not check the validity of the cross compensation of interferences, and we

rely on the specifications given by the producer. Usually the cross compensation for targeted gases is a routine work and could be done with the calibration process through a self-calibration or calibrated by the producer. On the manual of Innova 1312, there is a chapter to explain how the users can conduct a self-calibration after moving the instrument around.

13. Line 188 - 190. Is there a reason to assume that the time response of the PAS instrument depends on the magnitude of the ammonia concentration?

Response: Higher concentrations will reach saturation faster and therefore responses are quicker.

14. Line 191 - 207. The purpose of the experimental setup (Figure 1) is not entirely clear. Was is used as a quantitative source of VOC concentrations for calibration of the PTR-MS? If so, which method was applied to determine the VOC concentrations in the gas phase? Details should be given! Or was the setup simply used as a source of VOC/air mixtures and the VOC concentrations were measured by a calibrated PTR-MS. If this is case, how was the PTR-MS calibrated for the selected VOCs?

Response: In fact, Figure 1 was the experimental setup for '2.4    Experiment 3: Laboratory test for correction factors', not for the selection of VOCs in '2.3    Experiment 2: VOCs selection test'. For the selection of VOCs, a clean plastic container which contained half-filled silage was used, with the air dragged by the PTR-MS. For the laboratory test for correction factors, a setup showed in Figure 1 was simply used as a source of single VOC+air mixture and the specific VOC concentration was measured by the PTR-MS which was calibrated as discussed above. The uncertainty is mainly coming from the transmission and reaction rate estimation, which is around 10-15% (Cappellin et al., 2012).

15. Line 206 - 207. The selected components (methanol, ethanol etc.) should be mentioned with reference to Section 3.2.

Response: As we thought that the selection of VOCs were the part of results for section 3.2, therefore we did not include the specific list of selected components in the materials and methods section 2.3. Now we have mentioned the selected components in the revised manuscript, please see Line 223-225.

16. Line 212 - 220. What is the meaning of a "pre-tested water solution" (Line 213) ? What were the concentrations of the liquid solutions and their temperature? How were the VOC concentrations in the gas phase quantified? How stable were the gas-phase concentrations? How large was the water vapor concentration in the diluted gas which was fed into the instruments?

Response: "pre-tested" was now removed in order to clarify the text. The water solution was prepared by using a volume ratio of VOC:Water as 1:5, with purging by clean air controlled by

2 mass flow controllers in order to reach a desired range for test (See Line 240-241). The temperature was not controlled and was assume to be room temperature. The VOC concentration was quantified by the calibrated PTR-MS. The water vapor concentration in the diluted gas which was fed into the instruments (not for PTR-MS) was typically from a few hundred ppbv up to 50 ppmv (as can be seen from Fig.S1). For the PTR-MS, a further dilution was applied, when the VOC concentration was higher than 10 ppmv.

17. Line 221 - 224. What is the meaning and function of "excess flow". The excess flow should be indicated in Figure 1.

Response: The excess flow is the extra flow required by the instrument of Innova and Picarro, in order to keep the right pressure in the instrument. The two arrows near PAS and CRDS in Figure 1 indicate the excess flow, where the exhaust line is added as suggested.

18. Line 236. Which company has manufactured the Multiplexer 1309? How does it work and what are the materials that come into contact with the sampled air?

Response: . The Multiplexer 1309 was also manufactured by the Danish company Lumasense Technology A/S, and now the information has been added as shown in Line 243-244. This product now has a new version as 'Multipoint Sampler - INNOVA 1409' (https://www.lumasenseinc.com/EN/products/gas-sensing/innova-gas-monitoring/photoacoustic-spectroscopy-pas/multipoint-sampler-1409/multipoint-sampler-innova-1409.html). The materials that come into contact with the sampled air is PTFE.

19. Line 244. Which "selected VOCs" and "odorants" were measured? The term "odorant" should be defined.

Response: s. The 'selected VOCs' are now clarified as 'all VOCs showed in section 2.3 were included', while the term of 'odorant' is defined and the selection were also given. Please see Line 264-265 in the revised manuscript.

20. Line 245 - 246. What is meant by "background"? Where were the four sampling lines going to?

Response: The four sampling lines going to the four locations with 'two selected locations inside the farm, one location in the pit ventilation, one location outside the farm.'. The background site was selected as the outside air beside the trailer, where the instruments were standing. This is now clarified in the revised manuscript, please see Line 267-268 in the revised manuscript.

21. Line 251 - 254. For which compounds were calibrations performed? The suppliers of the calibration gas mixtures and permeation device should be mentioned. What is the accuracy of the calibrations?

Response: Permeation tubes (VICI Metronics, Inc., Houston, TX, USA) included acetic acid, propanoic acid, butanoic acid, pentanoic acid and 4-methylphenol. Gas mixtures (all 5 ppmv in nitrogen) included hydrogen sulfide (AGA, Copenhagen, Denmark), methanethiol (AGA, Copenhagen, Denmark), and dimethyl sulfide (Air Liquide, Horsens, Denmark). Details regarding the calibration procedures could be found in our previous study, with accuracy with 12% error and in most cases within 8% (Liu et al., 2018). This is now clarified in the revised manuscript, please see Line 275-281.

22. Line 281 - 282. "Background concentrations of ammonia measured..." can be misunderstood and should be rephrased. Do you mean instrumental baseline (instrumental offset) when ammonia-free zero air is measured? Is there a plausible explanation for the background values of the CRDS and PAS instruments?

Response: Yes, we meant the instrumental baseline when ammonia-free zero air was measured. Now the sentence is revised in order to avoid misunderstanding. For PAS, the baseline is probably due to water vapor interference. For CRDS the baseline is really low (1 ppbv), since low ppbv concentrations are present more or less everywhere, e.g. from human breath.

23. Line 292 - 296, Figure 2B. The result of the instrumental comparison needs more discussion. Are the calibrations of the two instruments (CRDS, PTR-MS) independent? What is the statistical error of the slope of the linear fit? Are the differences between the two instruments statistically significant and can they be explained by instrumental calibration errors? The measurement comparison between PAS and CRDS (or PTR-MS) should also be shown and discussed.

Response: The ammonia concentration was simultaneously measured by the CRDS and the PTR-MS for Figure 2B. The SD of the slope of the linear fit was 0.005 which is really small. Therefore, the error is not much, and probably the slope is not significantly different from 1. The simultaneously measurement (calibration and comparison) between PAS and CRDS (or PTR-MS) for Ammonia was not performed in this study, but should be investigated in the future.

24. Line 300 - 315. What could be the reason for the concentration dependence of the PAS response time? Why are the PAS values elevated at 18:43 (Fig. 2C) and decrease until 19:00, while CRDS and PTR-MS show constant values. After the ammonia concentration has been switched to zero, the decay of the PAS signal seems to have at least two time constants. There is an initial fast decay that is followed by a tail with a slow decay. What could be the reason for the time behaviour? How long does it take until the PAS signal reaches baseline values?

Response: The three instruments were only used at the same time from around 19.00, and data before this time were from another test. Therefore, showing of the PAS values at 18.43 in the figure was not appropriate. Now we have corrected this error in the revised Figure.2C. We don't have an explanation for the time behavior (two time constants), which may be included in the future study. It took around half an hour for the PAS signal reaches baseline values.

25. Line 378 - 380. Interferences in ammonia PAS measurements have been studied for single VOCs in air. Have you tested whether the interferences are additive in a multicomponent gas mixture? Additivity would at least require that the optical absorbances of the interfering VOCs are small (avoiding line saturation). (Non)linearity of the relationships seen in Figure 4 may give useful information. For a mixture with multiple interfering VOCs, a valid correction using the information from Table 2 can only be expected, if the interferences are independent of each other. This aspect needs to be discussed.

Response: Only single VOC was tested in the lab regarding interreference on PAS and correction factors were obtained for single VOC. We have to acknowledge that we don't know if the effects are always additive – although they are likely to be. In the field study, ethanol certainly dominated the VOC matrix in general, but other types of VOC also contribute significantly. For example, the averaged ratio of ethanol concentration to the sum of the 8 VOCs (tested in lab with obtained correction factors) was 0.64 ($\pm$0.11) for Location Two in the field study. From this single application it seemed that the obtained empirical relationships to be additive, but a complete investigation is indeed needed in the future. This part is now revised accordingly and please see Line 553-560.

26. Table 2. For which concentration ranges were the relationships tested? This information should be included in the table. Is there a physical explanation for negative interferences?

Response: . The concentration range indeed should be clarified, and now it is added in the text for Table 2. The negative interferences can usually be explained by the internal cross compensation procedure for one target filter (first target filter, such as $NH_3$ filter) on positive artifacts at another target filter (second target filter, such as $CH_4$ filter) caused by non-target gas (such as VOC) on the second target filter. This physical explanation was included in a few relevant references such as Zhao et al. (2012).

27. Line 412 - 425. Is there a physical explanation for the nonlinear behaviour of the VOC interference in measurements of $N_2O$? Are the nonlinear interferences additive when two or more interfering VOCs are present in measured air?

Response: . We do not have physical explanation for the nonlinear behavior of the VOC interference in measurements of $N_2O$, and we also don't know if these nonlinear interferences are additive or not when multiple VOC presented. In the field measurement, we could not determine the $N_2O$ concentration by other instrument than PAS, and therefore could not determine the interference for $N_2O$ by VOCs. In the future, this might be worth a sophisticated investigation just for $N_2O$ interference.

28. In Figure 2B, 4, 5B, 5D, S2, S3 and Table 2, results from statistical data treatments are shown. More information should be given on how fitted lines (curves) and fitted parameters were obtained. Were measurements corrected for offsets (background)? How were linear fits

obtained (least square fits? with or without error weighting?). Is it justified to force the fit through the origin? Fitted parameters should be given with $1\delta$ errors. The meaning of the plotted error bars in Figure 4 should be explained. The error bars (statistical errors?) are significantly larger than the scatter of the data points around the fitted line. What does that mean?

Response: All measurements were corrected from background and the linear fits were least square fits without error weighting. From the figures, fitting equations were given. The fitting equations given that the fits were forced through the origin(zero). In the revised manuscript, this is now clarified for the mentioned Figures and table. Fitted parameters are given with $1\delta$ errors, as shown in Table 2. The plotted error bars in Figure 4 were representing the standard deviations for the measured VOC by the PTR-MS under a selected VOC level (x-axis) and for the measured NH3/N2O level by the PAS meanwhile (y-axis). This is now in the text for Figure 4. The error bars are statistical errors, which are generally within reasonable range.

Technical comments

1. Materials and methods. More information about used gases (nitrogen, zero air), chemicals

(organic compounds listed in Table 2) and the water used for VOC solutions should be given.

Suppliers and purity grades need to be specified. How was zero air generated? How were the

solutions of VOCs in water prepared?

Response: . The zero air was supplied from a HiQ zero air station (Linde AG, Munich, Germany). Nitrogen and clean air were supplied through a charcoal/silica gel filter. The 8 selected VOCs were purchased from Sigma-Aldrich with at least analytical grade purity. The water solution was prepared by using a volume ratio of VOC:Water as 1:5, with purging by clean air controlled by 2 mass flow controllers in order to reach a desired range for test.The manuscript was revised accordingly (Line 240-241).

2. Line 34. The term "field study" may be confused with a study under natural ambient conditions. Change sentence to "measured by PAS in a dairy farm".

Response: . "from a field study" is now revised to "in a dairy farm" as suggested.

3. Line 41 - 42. Ammonia causes soil acidification?

Response: . With the nitrification process the ammonia transfer to nitrate and hydrogen into the soil and increase the soil acidification.

4. Line 67 - 70. PAS is a general term for a spectroscopic method, but here you refer to a particular instrument (Innova 1312). Therefore, it is better to say: "Besides, the Innova 1312

has the advantages ...".

Response: . We have revised according to the suggestion.

5. Line 69. Change to "Usually, water vapor is also measured in order ...": Changed now.

6. Line 73. Change to "absorption of infrared light": Changed now.

7. Line 80. There is a word missing in "interference of has not been well studied...": "of" is now deleted.

8. Line 81. "Mathot et al., 2007" is missing in the Reference section.: This reference is now added to the Reference section.

9. Line 118-199: Change to "were used to measure trace gas concentrations in air".: Changed now.

10. Line 125 - 127. The sentence is not clear and should be rephrased.: The sentence is now rephrased.

11. Line 141. Either "acceleration of the ring down " or "reduction of ring down time ".: This is now revised.

12. Line 143. "Normal" ring down time needs to be explained.: This is now explained.

13. Line 173. Change to "Instrumental background signals, ammonia calibrations and instrumental time responses were characterized for the PAS, PTR-MS and CRDS instruments".: Changed now.

14. Line 180. Delete "for the calibration test".: Deleted now.

15. Line 181. The desired concentration range should be specified.: Specified now.

16. Line 197. Which kind of plastic material was used for the container? Was it emission free?

Response: Teflon plastic was used and it was emission free. Manuscript was revised accordingly.

17. Line 254. A reference for "Standard conditions as described previously ..." should be provided.: Reference is given now.

18. Line 293. Change to "in which the slope of the fitted line (k = 0.96 ± ?) ..." It's changed now.

19. Line 304. "90% decay time" needs to be defined. It's defined now.

20. Line 380. Change to "...VOC concentrations be measured simultaneously by other

instruments". It is revised as "VOC concentrations be measured simultaneously by expensive analyzers as PTR-MS"

21. Line 382. The term "correction factor" should be defined. It is now revised to empirical relationship (ER) as suggested in another comment to Table 2.

22. Figure 1. The exhaust line should be marked.: Exhaust line is now marked.

23. Figure 2. All axis should show tic marks. Concentrations at the x and y axis of Figure 2B and at the y axis of Figure 2C should be given in ppmv.

Figure 2C caption: change to "Instrumental response of PTR-MS, PAS and CRDS instruments to a rectangular ammonia concentration pulse."

Figure 2D caption: change to "Instrumental response of PTR-MS, PAS and CRDS instruments to a stepwise increase in ammonia concentration."

Response: Figure 2 is now revised as suggested. Figure 2D caption changed to "Instrumental response of PAS instrument to a stepwise increase in ammonia concentration."

24. Figure 4. All axis should show tic marks. Concentrations at the x and y axis of Figure 4A and 4B should be given in ppmv. The Figure caption mentions red, green and purple lines. The colour designation should be made consistent with the plotted lines.

Response: Figure 4 is now revised as suggested.

25. Figure 5. All axis should show tic marks. Concentrations at the x and y axis of Figure 5B and 5D should be given in ppmv. Draw 1:1 lines in A and C as reference.

Response: It's been revised as suggested.

26. Figure S1. Concentrations should be given in ppmv.: It's been revised.

27. Figure S2. All axis should show tic marks. "ppm" should be "ppmv".

28. Figure S3. All axis should show tic marks. Concentrations should be given in ppmv. What is the difference between the upper and lower panel in Figure S3? It's been revised. The difference is Location One and Location Two for the up and down panel, respectively.

29. Table 1. Units should be ppmv (to be consistent with text and figures).

Response: Table 1 is now revised. For the detection limit, we kept ppbv since the values are basically within lower range of ppbv.

30. Table 2 caption must be rephrased. The table does not show correction factors, but empirical relationships describing the functional dependence of the interference in the measurement

of the target compound (e.g., NH3) on VOC concentrations.

Response: Table 2 is revised as suggested.

31. Table 3. Do standard deviations apply to the mean values, or do they represent the variability of single measurements? The concentration values for ethanol and methanol should be reasonably rounded.

Response: Yes the standard deviation apply to the mean values. It's now clarified in the revised manuscript.

32. Table S1. "Filter center" should be changed to "Center wavelength". What is the meaning of "Band width (%)" ? Is it related to "Filter bandpass"?

Response: "Filter center" is now changed to "Center wavelength" as suggested. Yes the "Band width (%)" is related to "Filter bandpass" which is calculated based on the center wavelength and band width (e.g., 2171=2215-2215*2.0/100 for UA0985).

Reference:

Cappellin, L., Karl, T., Probst, M., Ismailova, O.,Winkler, P.M., Soukoulis, C., Aprea, E., Märk, T.D., Gasperi, F., Biasioli, F., 2012. On quantitative determination of volatile organic compound concentrations using proton transfer reaction time-of-flight mass spectrometry. Environ. Sci. Technol. 46, 2283–2290.

Kamp, J.N., Chowdhury, A., Adamsen, A.P.S., Feilberg, A.: Negligible influence of livestock contaminants and sampling system on ammonia measurements with cavity ring-down spectroscopy. Atmos. Meas. Tech., 12, 2837–2850, https://doi.org/10.5194/amt-12-2837-2019, 2019.

Zhao, Y., Pan, Y., Rutherford, J., and Mitloehner, F. M.: Estimation of the Interference in Multi-Gas Measurements Using Infrared Photoacoustic Analyzers, Atmos., 3, 246–265, https://doi.org/10.3390/atmos3020246, 2012.

---

## Author Comment (AC3) · 27 Sep 2019

The comment was uploaded in the form of a supplement:
https://www.atmos-meas-tech-discuss.net/amt-2018-412/amt-2018-412-AC3-supplement.pdf

---

## Author Comment (AC4) · 27 Sep 2019

Response to SC2:

We were always surprised how confident researchers have been in trace gas instruments using broadband light sources for the determination of gaseous emissions from agriculture. The present paper focuses on interferences occurring in ammonia concentration determination by INNOVA instruments. It convincingly shows the severe limitations of this measurement technology and documents the strong influence of other gases on the readout of these devices. Reliable concentration measurements need in addition the determination of a large variety of compounds in order to perform the suggested correction algorithms. This is a paradoxical situation, as such measurements need complex and costly instrumentation that will make the use of broadband filter based instruments redundant. In some cases cheaper NH3 passive samplers or liquid impingers can be used as a robust and reliable alternative, if high time resolution is not required. Innova-based NH3 concentration measurements are likely influenced by compensating errors. The positive interferences discussed in this paper focus on the detection side, but in parallel there are potential losses of NH3 in the inlet tubing/lines and switching valves, especially in multiport systems. Within livestock production systems rather high dew points are present that facilitate absorption onto most types of surfaces. Often low flow rates are used, tubing is not heated, and some inlet filters are put in place to protect the analyzer from dust ingress, which all exacerbate the adsorption/desorption problem.

It is also striking that in many peer-reviewed articles reporting emission data based on Innova or similar instruments, detailed information on the sampling and analytical system are missing, which prevents a critical re-evaluation of such data. Consequently, emission factors based on these instruments should be taken with great caution.

Fifteen years ago, we investigated in detail the dependence of the concentration output, for a large range of CO2-CH4-N2O-H2O mixtures, of three different Innova's (1312, 1314) as a function of the cell temperature and the water vapor (dew point). An overview of the measurements is given in the supplement. These analyses are the base of the evaluation algorithm developed by Flechard et al. (2005) for the determination of N2O concentrations, which showed the large influence of CO2, water vapour and temperature. Another striking result was the large influence of the water content and the cell temperature on the CH4 concentration. The contribution of H2O to the CH4 raw signal depends on the ratio of the two gases. It will e.g. be less important in cow stable, compared to our INNOVA experiment where CH4 was only a couple of ppm.

Only in case Innova devices are operated in a temperature-controlled environment, with the dew point being kept constant, can reliable CH4 measurements be made.

Reference Flechard, C. R., Neftel, A., Jocher, M., Ammann, C., and Fuhrer, J.: Bidirectional soil-atmosphere N2O exchange over two mown grassland systems with contrasting management practices, Glob. Change Biol., 11, 2114–2127, 2005.

Albrecht Neftel Neftel Research Expertise Christoph Flechard INRA, Rennes

Please also note the supplement to this comment:

https://www.atmos-meas-tech-discuss.net/amt-2018-412/amt-2018-412-SC2-supplement.pdf

Response: Thanks for the comment. We agree that reliable concentration measurements need in addition the determination of a large variety of compounds in order to perform the suggested correction algorithms. Nevertheless, depending on the application case, the dominating compounds might be limited to a few major ones, where the measurements and correction likely be possible. Still, we agree that such measurements generally need complex and costly instrumentation which make the use of broadband filter-based instruments less attractive.

In this study, we focused on interference for the detection side, there are indeed some potential losses of NH3 in the inlet tubing/lines and switching valves, which we did not investigate in this study. Typically, the PTFE tubing is used, therefore the adsorption/desorption problem was not focused but need systematic investigation in the future.

We agree very well that we have to take great care when using emission data from many peer-reviewed articles based on Innova or similar instruments, especially when detailed information on the sampling and analytical system are missing, and the corresponding emission factors should not be used otherwise.

We admit that we did not look into the effect of temperature which seem to influence significantly the greenhouse gas concentration measurement especially memthane. In the future this should be investigated together with combined interference effects from various compounds on ammonia and greenhouse gas measurement by the broadband light PAS.

---

## Author Response (AR2)

Response to Minor revision:

The following technical changes should be applied.

The statement about the negative interferences given in the answer to Comment 26 should be explicitly included in the paper.

Response: The statement about the negative interferences given in the answer to Comment 26 is now included in the revised manuscript (see Line 443-448 in the revised manuscript).

I suggest to modify the caption of Figure 2 as follows:

Figure 2. Ammonia test measurements by PAS, PTR-MS and CRDS. A: Background signals measured in ammonia-free air. B: Intercomparison of ammonia concentrations measured by PTR-MS and CRDS. C. Instrumental response ....

Response: The caption of Figure 2 is now revised as suggested. Please see Line 299-301 in the revised manuscript.

The abbreviation "SD" should be defined (SD = standard deviation ?).

Response: The abbreviation "SD" is now defined in the manuscript when it showed first time in Table 1 (see Line 331-332).

Table 2: Up limit --> Upper limit

Response: It is now revised as suggested. Please see Table 1.

Table 2: I suggest to change the text of the footnote A as follows:

"Not specified by the producer".

Response: It is now revised as suggested. Please see Line 337.

Table 2: the column "accuracy" should specify the $1\sigma$ accuracy of the ammonia concentration data measured by the instrument. This is usually determined by the total error of calibration.

Response: "$1\sigma$ accuracy" is now specified. Please see Table 1.

Table 1 and 3: the number of significant figures (digits) should be adjusted to the precision of data.

Response: The number of significant figures are now adjusted in Table 1 and Table 3.

[revised manuscript text omitted]